# A hyperspectral method to assay the microphysiological fates of nanomaterials in histological samples

Elliott D SoRelle[1,2,3,4†], Orly Liba[1,2,4,5†], Jos L Campbell[1,6‡], Roopa Dalal[7], Cristina L Zavaleta[1,6], Adam de la Zerda[1,2,3,4,5*]

[1]Molecular Imaging Program at Stanford, Stanford University, Stanford, United States; [2]Bio-X Program, Stanford University, Stanford, United States; [3]Biophysics Program, Stanford University, Stanford, United States; [4]Department of Structural Biology, Stanford University, Stanford, United States; [5]Department of Electrical Engineering, Stanford University, Stanford, United States; [6]Department of Radiology, Stanford University, Stanford, United States; [7]Department of Ophthalmology, Stanford University, Stanford, United States

*For correspondence: adlz@stanford.edu

†These authors contributed equally to this work

Present address: ‡Ian Potter NanoBioSensing Facility, NanoBiotechnology Research Laboratory, School of Science, RMIT University, Melbourne, Australia

Competing interests: The authors declare that no competing interests exist.

**Abstract** Nanoparticles are used extensively as biomedical imaging probes and potential therapeutic agents. As new particles are developed and tested *in vivo*, it is critical to characterize their biodistribution profiles. We demonstrate a new method that uses adaptive algorithms for the analysis of hyperspectral dark-field images to study the interactions between tissues and administered nanoparticles. This non-destructive technique quantitatively identifies particles in *ex vivo* tissue sections and enables detailed observations of accumulation patterns arising from organ-specific clearance mechanisms, particle size, and the molecular specificity of nanoparticle surface coatings. Unlike nanoparticle uptake studies with electron microscopy, this method is tractable for imaging large fields of view. Adaptive hyperspectral image analysis achieves excellent detection sensitivity and specificity and is capable of identifying single nanoparticles. Using this method, we collected the first data on the sub-organ distribution of several types of gold nanoparticles in mice and observed localization patterns in tumors.

## Introduction

Nanoparticles (NPs) can be fashioned in precise shapes and sizes from a wide variety of materials. This synthetic versatility makes NPs excellent tools for wide-ranging biomedical applications including in vivo imaging (*Jokerst et al., 2012*; *Durr et al., 2007*), drug delivery (*Hauck et al., 2008*), photothermal therapy (*Huang et al., 2006*; *von Maltzahn et al., 2009*), and gene transfection (*Huang et al., 2009*). In particular, metal and metal oxide NPs made from gold, silver, iron, and titanium are commonly used in biomedicine owing to their unique electromagnetic properties (*Giustini et al., 2011*; *Husain et al., 2015*; *Lee and El-Sayed, 2006*). Once administered to a living subject, these NPs may exhibit vastly different pharmacokinetics and uptake profiles that are contingent on NP shape, size, surface coating, and other factors (*Owens III and Peppas, 2006*; *Chen et al., 2009*; *Zhang et al., 2009*; *He et al., 2010*; *Decuzzi et al., 2010*). These differences manifest not only at the scale of whole organs but also at the cellular level (*Giustini et al., 2011*; *Yang et al., 2014*; *Sadauskas et al., 2009*). Ideally, biodistribution studies should address various scales – from whole animal to tissue and cellular interactions – in order to understand a given NP's *in vivo* behavior.

**eLife digest** Metallic elements like gold and silver can be made into particles that are one thousand times smaller than the width of a human hair. Researchers can create these "nanoparticles" in different sizes and shapes that exhibit unique properties. For example, gold can be made into rod-shaped particles that interact with infrared light. Other nanoparticles can be loaded with drug molecules and designed to bind to cancer cells. As a result, nanoparticles have been explored for use in a variety of biomedical imaging and therapy applications. However, we must fully understand how the nanoparticles bind to the cancer cells and how the body tolerates these nanoparticles before they can be used in humans.

Experiments that explore where nanoparticles accumulate in the body are typically called biodistribution studies. However, current techniques for studying biodistribution cannot simultaneously measure the uptake of particles into organs and reveal the fine structures inside the organs that interact with the particles.

SoRelle, Liba et al. aimed to address this problem by developing a new biodistribution technique called HSM-AD (short for hyperspectral microscopy with adaptive detection). This new technique combines a relatively recent method called hyperspectral dark-field microscopy, which can identify nanoparticles from their unique optical signatures, with versatile computer algorithms to detect nanoparticles.

HSM-AD is more sensitive than previously developed biodistribution techniques, and SoRelle, Liba et al. used it to produce highly detailed maps of nanoparticle uptake patterns in the organs of mice. These maps provide new insights into how cells and tissues in the body handle different nanoparticles. Moreover, HSM-AD was able to distinguish nanoparticles with unique shapes by their distinct optical signatures. Further experiments show that HSM-AD can reveal interactions between human tumor cells and nanoparticles specifically designed to target those cells.

HSM-AD will be a useful resource for researchers studying the effect of nanoparticles on the human body. Future studies will use this technique to explore which nanoparticles have the potential to be developed for medical uses.

Current studies commonly employ inductively-coupled plasma (ICP) techniques (*Niidome et al., 2006*) or electron microscopy (EM) (*Giustini et al., 2011*) to interrogate metallic NP biodistribution. However, each of these techniques has notable disadvantages. ICP can be coupled to mass spectrometry (MS) or atomic/optical emission spectrometry (AES/OES) to quantify the presence of a metallic species in tissues of interest with high sensitivity (~10 parts per billion); incidentally, detection of large metal NPs with ICP relies upon dissolving samples in strong acids. The need to dissolve NP-containing samples has severe drawbacks with respect to characterizing particle uptake including the complete loss of spatial insights such as NP distribution patterns within the given tissue. Moreover, the sample preparation itself can impede detection sensitivity, especially for small tissue samples and tissues with intrinsically low NP uptake, which must be diluted in acid. Conversely, EM studies provide exquisite high-resolution images of NP uptake by individual cells. Unfortunately, EM requires cumbersome sample preparation and acquires qualitative data over fields of view that are too small to be tractable for whole organ studies. Fluorescence (*Zhang et al., 2009*; *He et al., 2010*; *Poon et al., 2015*) and radioactivity (*Kreyling et al., 2015*; *Collingridge et al., 2003*) detection can also be used to assess NP biodistribution, however these techniques typically require the addition of a labeling moiety to the NP prior to *in vivo* use. Aside from the potential that labels may detach or even alter NP pharmacokinetic properties, whole-organ studies with these techniques can be impeded by poor spatial resolution.

Hyperspectral dark-field microscopy (HSM) is a technique that obtains scattered light spectra from a sample on a per-pixel basis (*Roth et al., 2015*). HSM is capable of identifying individual nanoparticles in pure solutions and cell culture by their intrinsic scattering spectra without the addition of a labeling molecule (*Fairbairn, 2013*; *Fairbairn et al., 2013*; *Patskovsky et al., 2015*). This approach may be ideal for detecting metallic nanoparticles with unique visible and near-infrared (NIR) spectral signatures. Unlike current methods that characterize NP biodistribution, HSM

simultaneously achieves diffraction-limited spatial resolution and excellent detection sensitivity without destroying the sample. HSM has been used to study NP uptake in cell culture (*Yang et al., 2014*; *Fairbairn, 2013*; *Fairbairn et al., 2013*; *Patskovsky et al., 2015*) and the induction of toxic effects in tissue (*Husain et al., 2015*), but its use for characterizing NP biodistribution has not yet been demonstrated due to several outstanding constraints. The primary limitation that has prevented HSM from being used in evaluating the biodistribution of NPs in tissue is the inability to accurately distinguish NPs from the background of tissue scattering. To abate this limitation, we use a modified dark-field microscope that uses oblique sample illumination to enable 150-fold brightness enhancement and ~15-fold better signal to noise ratio (SNR) than standard dark-field optics (*Badireddy et al., 2012*; *Zhang et al., 2015*). Another challenge with HSM detection stems from the reality that individual NPs within a given sample do not exhibit the exact same spectrum. Furthermore, the NP uptake within tissues inevitably results in a combination of the NP spectrum with tissue scattering, which can be spectrally diverse. Current approaches such as spectral angle mapping (*Roth et al., 2015*; *Kruse, 1993*; *De Carvalho and Meneses, 1999*; *Luc, 2005*; *Roth et al., 2015*) (originally developed for non-biological applications) and manual delineation (*Husain et al., 2015*; *Roth et al., 2015*) cannot adapt to these conditions and may yield high false positive and false negative detection rates. It has been observed that no HSM method to date has demonstrated robust capabilities for quantifying false positive rates or other diagnostic measures (*Roth et al., 2015*). Thus, HSM methods must be customized to address spectral mixing and diffraction effects as well as detection sensitivity and specificity if they are to be successfully used for microscopic analyses of complex biological samples.

Here, we demonstrate Hyperspectral Microscopy with Adaptive Detection (HSM-AD), the first HSM method based on adaptive clustering, as a viable alternative to current techniques for assessing whole-organ biodistribution and cellular uptake of NPs. In this study, we collected tissues of interest from mice that were injected with large gold nanorods (LGNRs) (*SoRelle et al., 2015*), gold nanoshells (Nanoshells), and silica-coated gold nanospheres (GNS@SiO$_2$), and we developed pre-processing and adaptive algorithms to identify NPs that accumulated in tissue sections based on their spectral signatures. The implementation of an adaptive classification algorithm for spectral classification extended HSM's single NP detection capabilities to tissue samples with negligible false-positive detection. HSM-AD was sufficiently robust for detecting NPs in images of different organ tissues and images acquired using variable illumination conditions. This approach may be preferable to conventional biodistribution assays for studies that simultaneously require quantification of relative NP uptake in various clearance organs and wide-field high-resolution images with histological detail.

## Results

### NP injection, tissue preparation, microscopy, and HSM-AD

LGNRs (~100 × 30 nm) exhibiting a near infrared plasmonic peak (*Figure 1a*) were synthesized, biofunctionalized, and administered to healthy and tumor-bearing nude (*Foxn1$^{nu/nu}$*) mice as previously reported (*SoRelle et al., 2015*; *Liba et al., 2016*). Mice were euthanized 24 hr post-injection, and various tissues were resected and fixed in 10% formalin. Fixed tissues were sectioned into 5 µm thick slices, mounted on glass slides, and stained with Hematoxylin and Eosin (H&E) as per standard histological preparation (*Figure 1b*). H&E-stained sections were imaged at 40x or 100x magnification in conventional dark-field and hyperspectral microscopy modes (CytoViva) (*Figure 1—figure supplement 1*). Conventional dark-field images (*Figure 1c*) were used to guide anatomical feature identification. All spectral data and quantitative comparisons presented in this report were derived from the analysis of hyperspectral images.

A hyperspectral camera with a detection range of 400–1000 nm was used to image scattered light from each sample in transmission mode. In the resulting images (*Figure 1d*), each pixel contains the spectral profile of the sample at the corresponding spatial position and can be used to detect LGNRs with near diffraction-limited resolution (1 µm). While standard dark-field images did not reveal notable differences between uninjected and injected samples, the hyperspectral images, in which three bands of the spectrum (800.0 nm, 700.6 nm, and 526.2 nm) were respectively color-coded as red, green, and blue, indicated that species with strong near infrared scattering (putative LGNRs, depicted in orange) were present in injected tissues but not observable in control tissues.

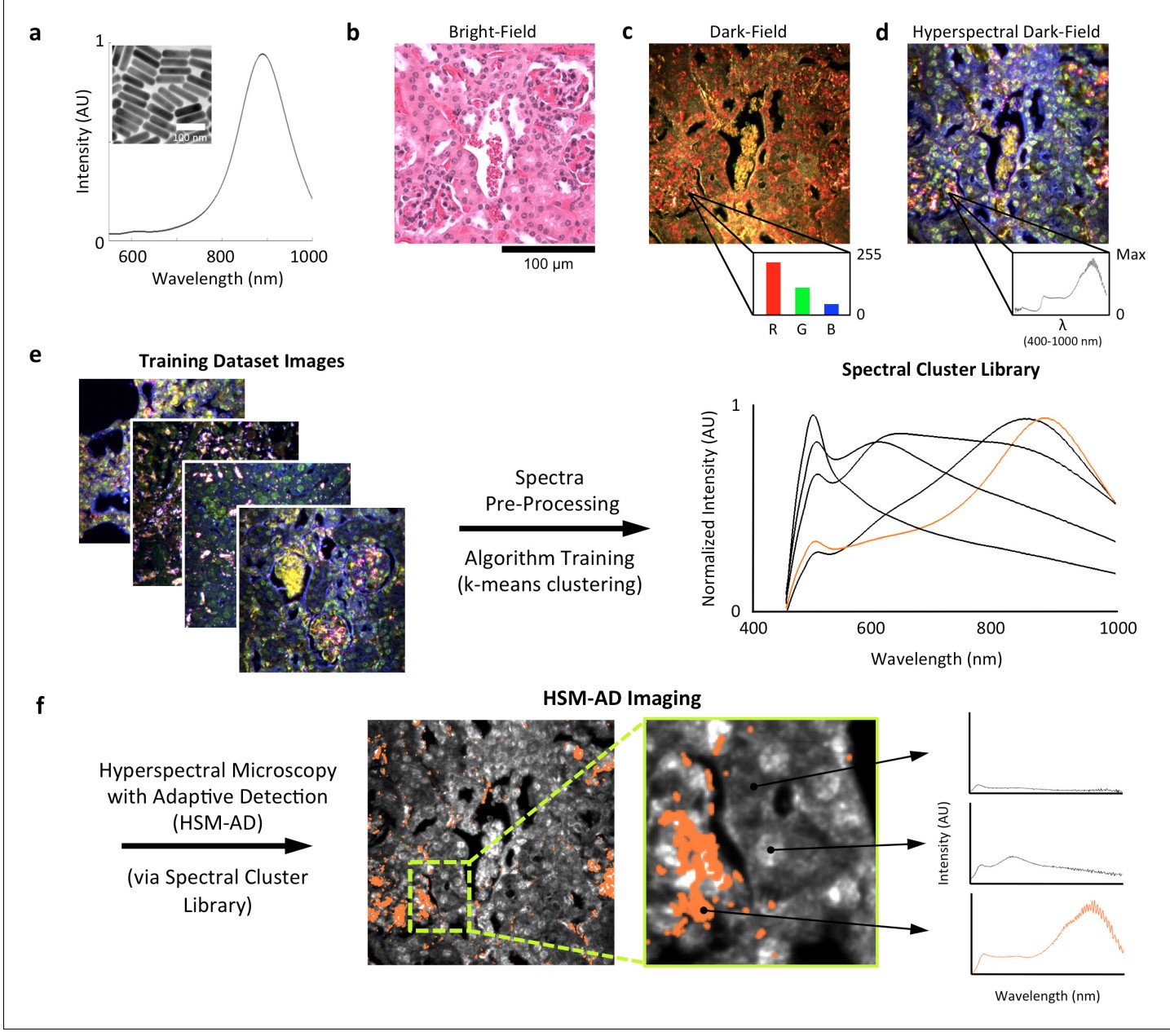

**Figure 1.** Overview of nanoparticle biodistribution analysis with HSM-AD. (**a**) Large gold nanorods (LGNRs, ~100 × 30 nm) exhibiting near infrared plasmon resonance were synthesized, functionalized, and intravenously injected into live nude mice. (**b,c**) 24 hr post-injection, the animals were euthanized and tissues were resected and prepared as normal histological sections for characterization with bright-field (**b**) and dark-field microscopy (**c**) neither of which was able to visualize the distribution of the LGNRs. (**d**) The same section was then imaged with hyperspectral microscopy, which showed clear signs of LGNRs accumulation (denoted by red hues) in various areas of the tissue and exhibited spectral peaks matching the LGNR plasmon resonance. (**e**) We then trained an adaptive clustering algorithm for spectral identification of LGNRs with hyperspectral images from injected mice. The algorithm identified several characteristic spectra representing the tissue and the H&E staining, as well as one unique spectrum representing the LGNRs (depicted in orange), altogether representing a library of 5 spectra. Once a spectral cluster library is produced from the training dataset, images of unknown tissue samples can by analyzed for the presence of LGNRs via automated classification. (**f**) The resulting HSM-AD images depict the location of all points within the sample that exhibit the LGNR spectrum (orange for LGNRs, grayscale for tissue).

The following figure supplements are available for figure 1:

**Figure supplement 1.** Diagram of the CytoViva microscope used for dark-field and hyperspectral image acquisition.

**Figure supplement 2.** Image segmentation, including method for dynamic threshold determination.

*Figure 1 continued on next page*

*Figure 1 continued*

**Figure supplement 3.** Detailed flowchart of steps used in HSM-AD algorithm.

**Figure supplement 4.** Typical cluster results for pixel classification in an image of tissues with injected LGNRs.

We created HSM-AD, a method that combines pre-processing and adaptive classification algorithms to automatically detect and quantify LGNRs in hyperspectral images. The pre-processing stage is described in detail in the methods section (*Figure 1—figure supplements 2,3*). One of the first stages of processing includes vignette correction and a determination of whether each pixel in the image belongs to one of three categories—background, tissue, or potential LGNR—based on its average intensity across the measured spectrum. Only high-intensity pixels that belong to the potential LGNR group are classified by the adaptive algorithm (*Figure 1—figure supplement 2*). For the training of the adaptive algorithm, pre-processed images of tissue samples from LGNR injected mice were input into a standard k-means clustering algorithm (*Bishop, 1995*). Four initial clusters were identified using this scheme and were used to produce a spectral cluster library (*Figure 1e*). Owing to the unique spectral profile of LGNR scattering, the particle spectrum was automatically recognized as a separate cluster by the k-means algorithm. Next, a fifth cluster was manually added to the spectral cluster library to account for edge artifacts caused by chromatic aberrations that were frequently falsely detected as LGNRs (*Figure 1—figure supplement 3*). The five spectra in this library were used as cluster centers for automatic detection of LGNRs in tissue sections using a nearest centroid (or nearest-neighbor) classifier. In this scheme, pixels that exhibited a spectrum that was closest (in a Euclidean sense) to the LGNR cluster center were identified as containing LGNRs (denoted as LGNR+). The rest of the pixels were classified as not containing LGNRs (denoted as LGNR-). Initial validation of the algorithm shows that regions of the image that were detected as LGNR+ indeed exhibited the characteristic plasmonic peak at around 900 nm while pixels identified as LGNR- did not (*Figure 1f*). The mean and standard deviation spectra of pixels classified into each cluster for a representative image also indicate the high fidelity of the algorithm (*Figure 1—figure supplement 4*).

## Characterization of sensitivity and specificity

We characterized the sensitivity and specificity of HSM-AD by three methods. First, we measured the false positive rate in uninjected tissue samples to obtain a specificity of 99.7% (*Figure 2—figure supplements 1–4*). The false positives, which also have a spectral peak near 900 nm, usually appear near the edges of the tissue section. We attribute this red-shift of the spectrum to chromatic aberrations, ostensibly due to the spectral dependence of the diffraction diameter (*Lipson and Lipson, 2010*). Next, we measured the false negatives in an image of LGNRs in mounting media (CytoSeal 60, Electron Microscopy Sciences) on a glass slide and obtained a detection sensitivity of 99.4%. We attributed the false negatives to LGNRs with hybridized surface plasmon resonances (*Funston et al., 2009*), which resulted in spectral scattering that was different from the distinct plasmonic resonance of single LGNRs (*Figure 2—figure supplements 5,6*). Because all training and test samples were mounted using the same media, spectral shifts due to local refractive environments did not contribute to false detection (*Figure 2—figure supplement 7*). Independently we also calculated specificity and sensitivity by analyzing LGNR-injected tissue samples (see *Methods*). We obtained a sensitivity of 89.5% and a specificity of 98.5% using this approach. The high sensitivity of the automated algorithm is further evident from its ability to detect single LGNRs, both on a glass slide and in injected tissue samples (*Figure 2—figure supplements 5,8*).

## NP biodistribution study

We demonstrated HSM-AD as a potential biodistribution technique by analyzing various tissues resected from mice (*Figure 2*). For quantitative measurements of LGNR uptake, we analyzed kidney tissue from uninjected (*Figure 2a*, *Figure 2—figure supplements 2a,3a,4a*) and injected (*Figure 2b*, *Figure 2—figure supplements 9a,10a,11a*) mice. Our analysis found a relative LGNR

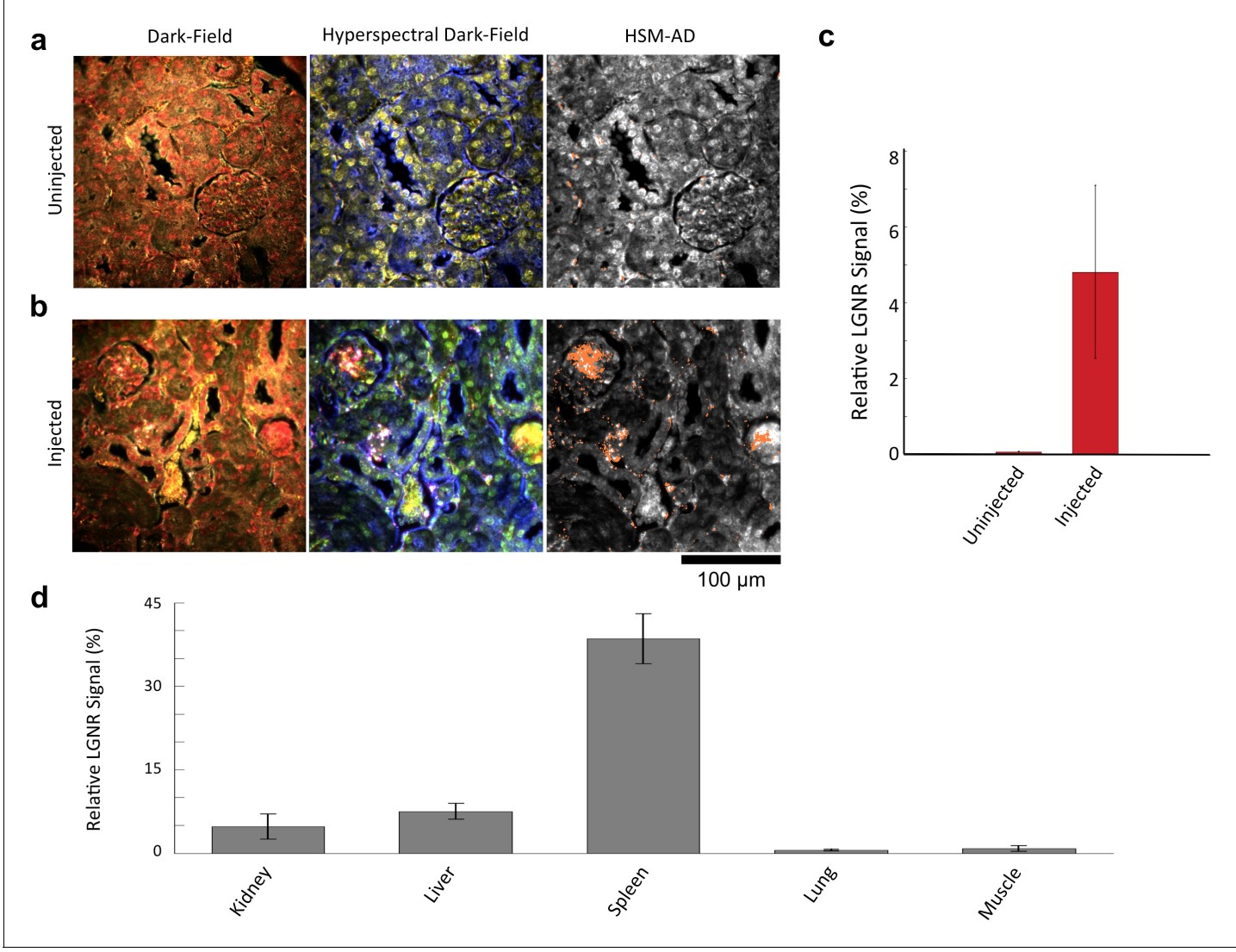

**Figure 2.** Sensitivity and specificity validation of HSM-AD. (**a,b**) Hematoxylin and Eosin (H&E) stained tissue samples (kidney) from uninjected (**a**) and injected (**b**) mice were imaged using dark-field and hyperspectral microscopy at 40x magnification and analyzed with HSM-AD to measure LGNR detection specificity and sensitivity. Conventional dark-field images highlight features including nuclei (salmon-pink), cytoplasm (green-brown), and erythrocytes (yellow-orange) within the tissues, but they reveal little information regarding the presence or absence of LGNRs. By comparison, putative LGNRs can be roughly identified as red-orange pixels in false-colored hyperspectral images, while nuclei and cytoplasm appear in green and indigo, respectively. (**c**) HSM-AD analysis of hyperspectral images demonstrates the absence of LGNRs in uninjected tissues and LGNR presence in injected samples (two-tailed Student's t-test, p=0.054). Quantification of the relative LGNR signal from n = 4 tissue slices (representing a total of 1.04 million pixels) indicates that the false positive rate for LGNR detection (determined from uninjected tissues) is minimal. A detection specificity of 99.7% was determined from uninjected tissue sections, and a detection sensitivity of 99.4% was measured from samples of pure LGNRs analyzed using HSM-AD (*Figure 2—figure supplement 1*). (**d**) HSM-AD analysis of whole tissue sections (n = 4 for each tissue type) reveals quantitative differences in bulk LGNR uptake among various organs, in a manner analogous to conventional biodistribution methods. Quantitative data are presented as mean ± standard error of the mean (s.e.m.).

The following source data and figure supplements are available for figure 2:

**Source data 1.** Data used for diagnostic and 95% CIs.

**Source data 2.** Data for whole organ uptake quantification.

**Figure supplement 1.** Measured sensitivity and specificity values for HSM-AD method.

**Figure supplement 2.** Dark-field images of additional uninjected H&E-stained tissue sections.

*Figure 2 continued on next page*

signal of 4.8% ± 2.3% in injected mouse kidney tissue. By comparison, a relative LGNR signal of 0.08% ± 0.01% was measured from uninjected samples, indicative of the method's high specificity (*Figure 2c*, *Figure 2—figure supplement 1*). Similar low false positive rates were measured in other organ tissues (*Figure 2—figure supplements 2b–e*,*3b–e*,*4b–e*). In addition to the kidney, HSM-AD was used to analyze LGNR uptake in liver, lung, muscle, and spleen sections to demonstrate an alternative to common biodistribution techniques. While a conventional biodistribution study of LGNRs has not yet been reported, HSM-AD analysis indicated that LGNRs exhibited a similar uptake profile (mostly in the liver and spleen) as commonly-used smaller gold nanorods (*Zhang et al., 2009*; *Niidome et al., 2006*). The greatest relative LGNR signal (38.5% ± 4.5%) was observed in the spleen. LGNRs were also concentrated in the liver (7.5% ± 1.5%). Particle uptake was minimal in lung tissue (0.5% ± 0.1%) and muscle tissue (0.8% ± 0.5%) sections (*Figure 2d*, *Figure 2—figure supplements 9–11*). Tissue sections without H&E staining were also analyzed and yielded results similar to those obtained for H&E stained sections (*Figure 2—figure supplements 12–14*).

## Sub-organ localization of LGNRs

HSM-AD imaging of histological sections (*Figure 3a*) enabled sensitive LGNR detection with sub-cellular resolution over large fields of view (*Figure 3b–c*), which afforded more detailed characterizations of NP uptake than those achieved by typical biodistribution methods. We used these advantages to investigate and quantify the sub-organ distribution of LGNRs. This analysis revealed well-defined patterns of LGNR uptake that appeared to be largely influenced by factors including particle size, innate immunological function, and waste-filtering anatomical structures.

The kidneys are responsible for filtering small, low molecular weight waste products from the bloodstream and diverting those products to the bladder for elimination (*Rouiller, 2014*). Waste-laden blood flows into capillary-dense structures called glomeruli within the kidney. Blood plasma containing small species including ions, biomolecules, cell fragments, and (in some cases) nanoparticles can extravasate from glomerular capillaries and traverse Bowman's space before being

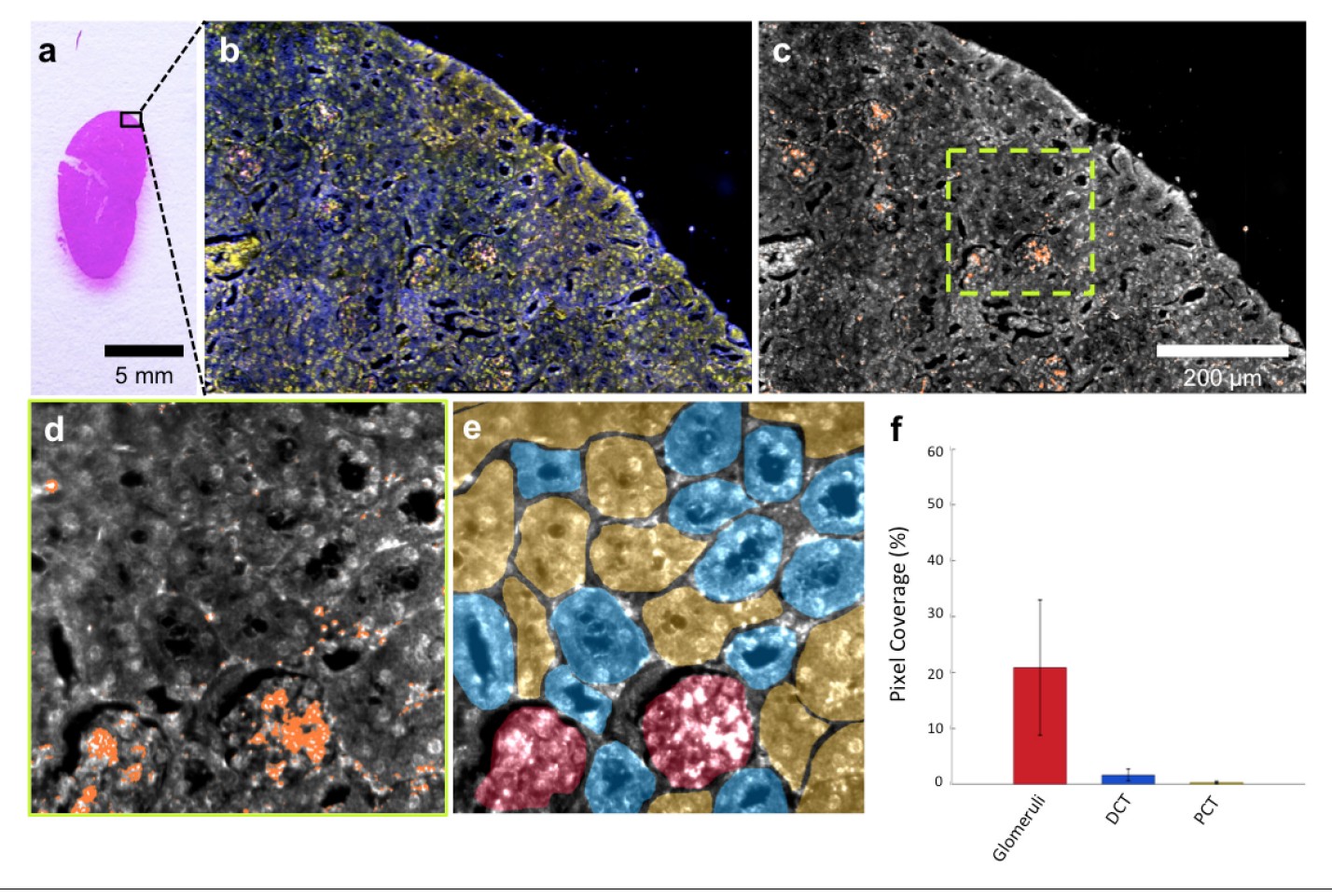

**Figure 3.** HSM-AD is capable of wide-field characterization of sub-organ distribution patterns of injected nanoparticles. (a–c) Millimeter-scale fields of view of histological sections of kidney. Photographed (a), acquired with near diffraction-limited resolution with a hyperspectral dark-field camera (b), and analyzed by HSM-AD (c) to reveal variable nanoparticle uptake within the fine anatomical structures. (d–f) As in conventional histology, micro-anatomical features of the kidney including glomeruli, Bowman's spaces, proximal convoluted tubule (PCT), and distal convoluted tubule (DCT) networks can be clearly identified in HSM-AD images (d). The ability to distinguish such histological details enables region of interest (ROI) analysis to quantify sub-organ accumulation of LGNRs (e,f). Quantification of the relative LGNR signal in glomeruli (red ROI), PCT (yellow ROI), and DCT (blue ROI) regions (e) revealed that the vast majority (~13-fold greater than in either tubule network) of renal LGNR uptake is localized within glomeruli (f). This is likely due to the size-dependent inability of LGNRs to traverse the ultrafiltration barrier formed by endothelial cells within glomerular capillaries. All quantitative data are represented as mean ± s.e.m. for each ROI type, as calculated from 4 unique fields of view acquired at 40x magnification.

The following source data is available for figure 3:

**Source data 1.** Data for kidney sub-organ ROIs.

collected into an extensive network of efferent tubes called Proximal Convoluted Tubules (PCT) and, further downstream, Distal Convoluted Tubules (DCT), which ultimately traffic waste from the kidney out to the bladder. Using our method, we observed that the vast majority of LGNRs within the kidney were concentrated in glomeruli and were virtually absent in the PCT and DCT, which are functionally downstream (*Figure 3d–f*). Glomerular uptake was 13-fold greater than uptake within the convoluted tubule network. These results can be explained by comparing the size of an individual LGNR with the narrow width of cellular junctions and the architecture of endothelial cells that form glomerular capillaries (*Satchell and Braet, 2009*). For reference, the sub-organ segmentation maps of all tissue images analyzed for quantification have been provided (*Figure 4—figure supplement 1*).

Along with the kidneys, the liver is instrumental in clearing waste from circulation. Partially because the size cutoff for hepatic filtration is notably larger than that of renal ultrafiltration (*Longmire et al., 2008*), we observed 1.6-fold greater LGNR uptake within the liver tissue compared to the kidney. Interestingly, the hepatic distribution of LGNRs also appeared to be non-uniform. Hepatocytes, which constitute the majority of liver tissue by mass, exhibited mild LGNR signal (2.9% ± 1.1%). By contrast, 15-fold more LGNR signal (43.5% ± 12.0%) in the liver was localized in a manner consistent with the shape, size, and number of Kupffer cells (*Figure 4a*, *Figure 2—figure supplements 9b*,*10b*,*11b*). We attribute this localization to the phagocytic function of Kupffer cells (*Owens III and Peppas, 2006*; *Longmire et al., 2008*). Thus, the variable localization of nanoparticles within the liver appears to be largely derived from the organ's innate immunological functions. It is interesting to note that aggregation within Kupffer cells likely caused spectral hybridization of some LGNRs, an observation that is consistent with previous studies of cellular uptake of gold NPs (*Chen, 2014*). This spectral shifting, which was most prevalent at the centers of LGNR aggregates, caused a portion of LGNRs to remain undetected by HSM-AD. While these aggregates were undetected by algorithmic means, their manual identification as LGNRs was evident from the lack of similar morphological features in uninjected liver tissue.

The spleen comprises several unique cell types arranged into tissues with diverse biological functions including blood filtration, innate immunity, and lymphocyte activation (*Mebius and Kraal, 2005*). The spleen is largely composed of red pulp, white pulp, and the boundary between these two tissues, commonly referred to as the marginal zone. Consistent with each tissue's biological function, we observed 1.7-fold greater relative LGNR pixel coverage in splenic red pulp than white pulp (*Figure 4b*, *Figure 2—figure supplements 9e*,*10e*,*11e*). A similar result has been previously reported for carbon-based nanomaterials (*Chen et al., 2015*). We did not definitively identify marginal zone tissue, but the radial distribution of LGNRs around white pulp follicles indicated that a significant portion of white pulp uptake may in fact be within the marginal zone (*Figure 4—figure supplement 2*).

Despite its dense network of alveolar capillaries, lung tissue exhibited minimal LGNR accumulation relative to the organs described above (*Figure 4c*, *Figure 2—figure supplement 9c*,*10c*,*11c*). This finding was consistent with existing biodistribution data for smaller particles, which can be explained by the lungs' major functions of gas exchange to and from the blood rather than biomolecule or particle filtration and clearance.

While whole-organ analysis indicated the presence of LGNRs in muscle tissue, HSM-AD revealed that muscle tissue itself (which consists largely of myocytes and dense networks of extracellular collagen) was virtually devoid of LGNRs (*Figure 4d*, *Figure 2—figure supplement 9d*,*10d*,*11d*). Rather, the apparently high LGNR presence was traced to blood vessels found in between muscle fiber bundles. As with the accumulation patterns described for other organs within this study, this distinction would not have been possible through conventional biodistribution methods.

HSM-AD images acquired at higher objective magnification (100x) offered further insights into the cellular nature of LGNR uptake within the kidney and liver tissue (*Figure 5*). Within the kidney, LGNRs were observed mostly within or in close proximity to glomerular capillaries (*Figure 5a,b*). HSM-AD also revealed patterns of LGNR uptake within individual Kupffer cells resident in liver sinusoids (*Figure 5c,d*). LGNR signal was detected within the Kupffer cell cytoplasm, but not within the region of the cell nucleus. This pattern is consistent with the phagocytic function of Kupffer cells in clearing particulate matter from circulation. Interestingly, several bright regions within the Kupffer cell were not detected as LGNRs. We expect that these regions resulted from spectral hybridization of LGNRs, possibly due to aggregation induced by lysosomal acidification following particle phagocytosis.

## HSM-AD detection and spectral unmixing of Nanoshells

We also injected mice with Nanoshells, which are morphologically distinct from LGNRs (*Figure 6a*). While Nanoshells and LGNRs both exhibit near-infrared plasmonic peaks, the Nanoshell spectrum is substantially broader than the LGNR spectrum (*Figure 6b*). A spectral cluster library was developed for H&E-stained Nanoshell+ tissues and was then used to quantify Nanoshell uptake as described for LGNRs (*Figure 6c*). While Nanoshells and LGNR displayed related uptake patterns, several differences including negligible Nanoshell uptake in kidney tissue and Nanoshell concentration within

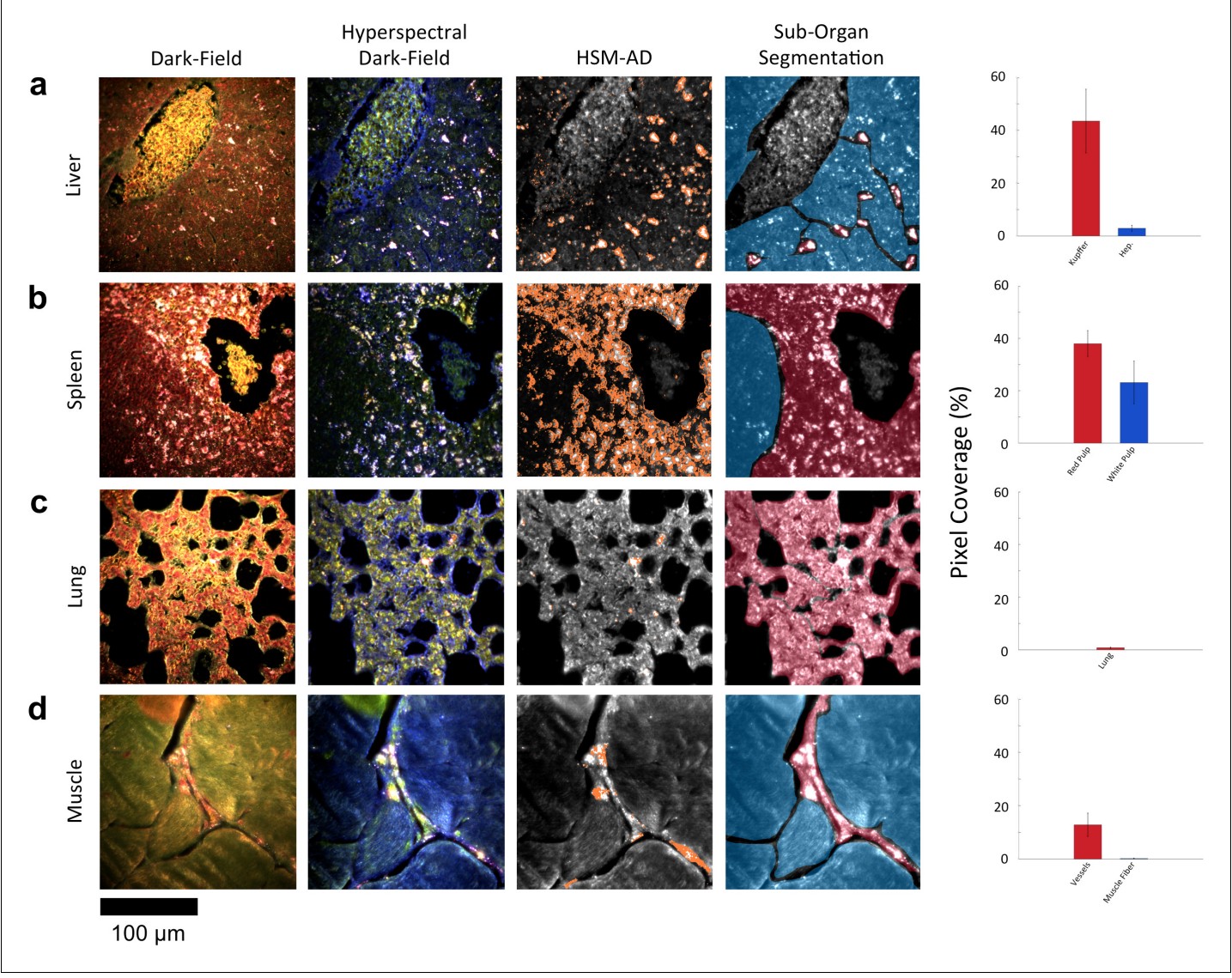

**Figure 4.** HSM-AD reveals characteristic patterns of nanoparticle microbiodistribution contingent upon tissue function. (**a**) LGNR accumulation in hepatic tissue occurs mostly in concentrated foci located within liver sinusoids. Along with the size, shape, and frequency of these foci, this pattern strongly suggests that these particles have been phagocytosed by Kupffer cells, the resident macrophages of the liver (red ROI). While there is mild uptake of LGNRs within liver hepatocytes (blue ROI), HSM-AD sub-organ quantification indicates that uptake by Kupffer cells is roughly 15-fold higher than hepatocytic accumulation. (**b**) The pattern of LGNR uptake in the spleen is also consistent with the physiological functions of various splenic tissues. A greater relative LGNR signal was observed in regions of splenic red pulp (red ROI), which is responsible for blood filtration, than in the white pulp follicles (blue ROI) that house B and T lymphocytes. (**c,d**) While LGNRs were prevalent within the liver and spleen tissues, HSM-AD results indicated minimal particle accumulation within the lung (**c**) or muscle (**d**) tissue samples (each < 1% relative LGNR signal for whole-tissue quantification). Interestingly, HSM-AD analysis demonstrated that the vast majority of LGNRs in muscle tissue sections were localized in blood vessels (red ROI) rather than within the muscle fiber tissue itself (blue ROI). Quantitative data are represented as mean ± s.e.m. as described previously.

The following source data and figure supplements are available for figure 4:

**Source data 1.** Data for liver, spleen, lung, and muscle sub-organ ROIs.

**Figure supplement 1.** Sub-organ region of interest (ROI) segmentation for additional tissue sections used for quantitative results presented in *Figures 3* and *4* of the main text.

**Figure supplement 2.** Detail of *Figure 4b*: spleen tissue histology correlated with LGNR uptake.

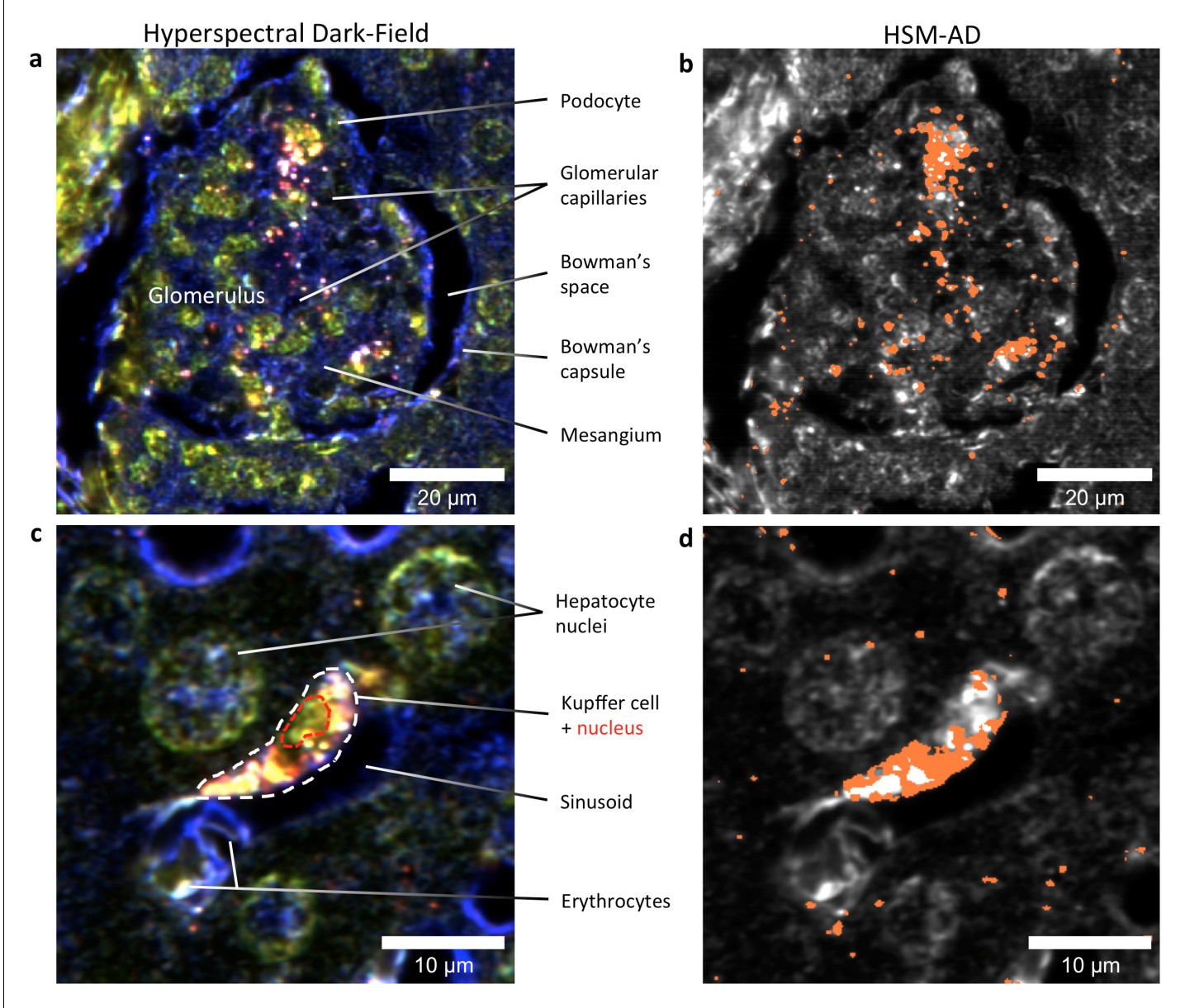

**Figure 5.** HSM-AD reveals the sub-cellular localization of intravenously administered nanoparticles with histological precision. (**a,b**) Hyperspectral (**a**) and HSM-AD (**b**) images of a renal glomerulus acquired at 100x magnification. A majority of LGNRS are found within or in close proximity to glomerular capillaries. Trace levels of LGNRs are observed in the kidney tissue outside of Bowman's capsule. (**c,d**) Zoomed views of Hyperspectral (**c**) and HSM-AD (**d**) images of liver tissue acquired at 100x magnification. Several erythrocytes and a Kupffer cell (dashed white line) can be observed residing within a liver sinusoidal vessel. Within the Kupffer cell, the nucleus (dashed red line) can be distinguished. HSM-AD analysis indicated the prevalence of LGNRs within the Kupffer cell relative to surrounding hepatocytes. The minimal LGNR signal was detected in the region identified as the nucleus, consistent with cytoplasmic LGNR localization. Several bright regions within the cell were not identified as LGNRs; these regions likely result from particle aggregation within acidic lysosomes following uptake by the Kupffer cell.

the splenic white pulp were observed (*Figure 6d*, *Figure 6—figure supplements 1,2*, Nanoshell+ pixels are shown in cyan).

HSM-AD was separately trained on a sample consisting of a mixture of pure Nanoshells and pure LGNRs. The spectral clusters identified during this training corresponded well to the spectra of each particle type and enabled high-specificity and high-sensitivity identification in samples of Nanoshells + LGNRs, Nanoshells-only, and LGNRs-only (*Figure 6—figure supplement 3*). These results

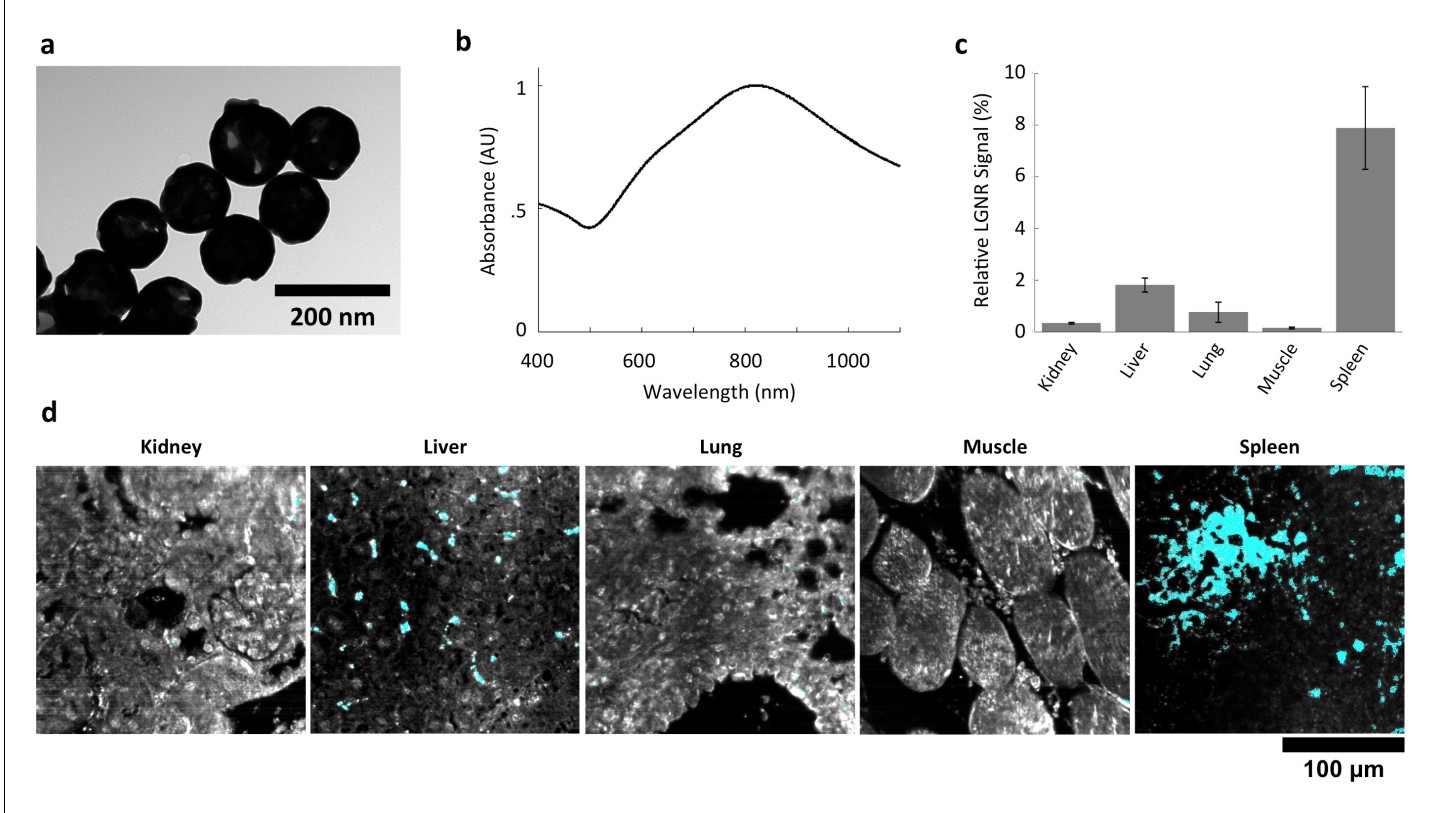

**Figure 6.** Characterization of gold nanoshell uptake after intravenous administration. (**a,b**) Nanoshells (119 nm silica core with 14 nm-thick gold coating) exhibit distinct particle morphology and composition (**a**) that (like LGNRs) yield a near-infrared (~800 nm) spectral peak (**b**). However, the Nanoshell spectrum is markedly broader than the resonance observed for LGNRs. (**c,d**) HSM-AD revealed that Nanoshell uptake displays inter-organ distribution patterns somewhat similar to those observed for LGNRs, with maximal accumulation in the spleen (**c**). Values are represented as mean ± s.e.m. from four FOVs per tissue. However, there are notable distinctions including minimal Nanoshell uptake within kidney tissue and concentration of Nanoshells within splenic white pulp (**d**) (Nanoshell+ pixels are depicted in cyan).

The following source data and figure supplements are available for figure 6:

**Source data 1.** Data for Nanoshell uptake in organs.
**Figure supplement 1.** Additional HSM-AD images of Nanoshell uptake used for quantification.
**Figure supplement 2.** Detail of Nanoshell uptake in spleen tissue.
**Figure supplement 3.** HSM-AD spectral unmixing of samples containing gold nanoshells and LGNRs.

demonstrate that HSM-AD can spectrally resolve plasmonic particles despite similarities in composition, although this capability was not tested in *ex vivo* tissues.

## GNS@SiO$_2$ detection in tissue

We used HSM-AD to characterize the tissue uptake of a third particle type, GNS@SiO$_2$ (*Figure 7*, GNS@SiO$_2$+ pixels are shown in green). Notably, GNS@SiO$_2$ are distinct from LGNRs and Nanoshells in terms of shape, size, composition, particle surface, and plasmonic resonance (*Figure 7— figure supplement 1a,b*). Because GNS@SiO$_2$ exhibit a visible regime plasmonic peak (~550 nm), HSM-AD analysis was performed on unstained tissue sections. First, a spectral cluster library was developed for GNS@SiO$_2$ classification as described for LGNRs (*Figure 7—figure supplement 1c*). Control tissues classified with this library displayed negligible false positives (*Figure 7a*). GNS@SiO$_2$ uptake in the liver and spleen was observed at 2 and 24 hr post-IV injection

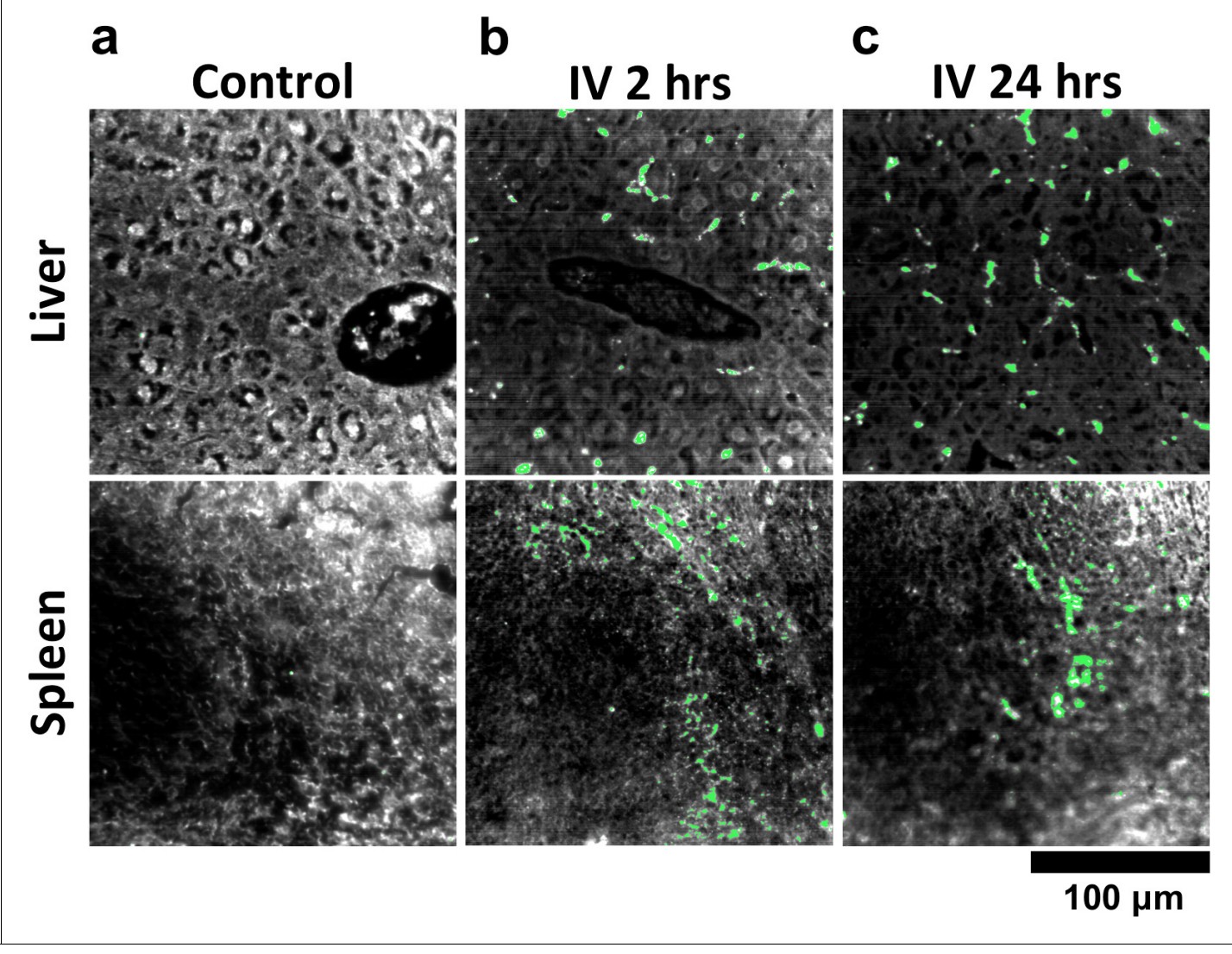

**Figure 7.** HSM-AD analysis of GNS@SiO$_2$. (**a**) Classification of unstained control tissues yields negligible false-positive detection for GNS@SiO$_2$ (green), which exhibit a peak plasmonic resonance of ~550 nm. (**b,c**) Intravenously-administered GNS@SiO$_2$ accumulate in the Kupffer cells of the liver and the marginal zone of the spleen within 2 hr (**b**) and persist up to 24 hr (**c**).

The following source data and figure supplements are available for figure 7:

**Source data 1.** Data for GNS@SiO$_2$ uptake in organs.

**Figure supplement 1.** Structural and spectral characterization of GNS@SiO$_2$.

**Figure supplement 2.** Detail of GNS@SiO$_2$ in spleen.

**Figure supplement 3.** HSM-AD quantification of GNS@SiO$_2$ uptake in liver and spleen tissue.

**Figure supplement 4.** Inductively-Coupled Plasma Mass Spectrometry (ICP-MS) quantification of GNS@SiO$_2$ uptake in liver and spleen tissue.

**Figure supplement 5.** Additional HSM-AD images of GNS@SiO2 uptake in liver tissue.

**Figure supplement 6.** Additional HSM-AD images of GNS@SiO2 uptake in spleen tissue.

(*Figure 7b,c*). Interestingly, GNS@SiO$_2$ uptake appeared to be even more localized to Kupffer cells than LGNR accumulation in the liver. Furthermore, GNS@SiO$_2$ in the spleen are consistently found in the marginal zone, and presence within the red pulp and white pulp is minimal (*Figure 7—figure supplement 2*). Quantitative results from HSM-AD correlate well with those obtained using ICP-MS (*Figure 7—figure supplements 3,4*), although it should be noted that HSM-AD measurements are more relative rather than absolute with respect to the amount of gold present in each tissue. As for LGNR quantification, four FOVs per sample were analyzed (*Figure 7—figure supplements 5,6*).

## Tumor uptake of targeted and untargeted NPs

One hallmark of tumor growth is angiogenesis, the stimulated development of new blood vessels to provide nutrients to rapidly dividing cancer cells. This newly-formed vasculature is composed of endothelial cells that express high levels of cell adhesion receptors including $\alpha_V\beta_3$ integrin (*Avraamides et al., 2008*). Thus, $\alpha_V\beta_3$ is commonly used as a target biomolecule for tumor imaging (*Sipkins et al., 1998*). Such studies have demonstrated that NPs targeted to $\alpha_V\beta_3$ exhibit greater accumulation in tumors in vivo than NPs coated with non-specific antibodies or small molecules. We hypothesized that the presence or absence of specific molecular targeting moieties would influence tissue-NP interactions beyond simply the degree of accumulation in target tissues. To test this, we used HSM-AD to observe the spatial patterns of targeted and non-targeted LGNR uptake within U87MG (human glioblastoma cells, $\alpha_V\beta_3^+$) tumor xenografts. We observed 7.4-fold greater relative LGNR signal of anti-$\alpha_V\beta_3$ LGNRs than isotype LGNRs in tumor tissue (*Figure 8a–d*). However, the most striking differences were in the localization patterns of each LGNR type. Anti-$\alpha_V\beta_3$ LGNRs were present in high density around the edges of small blood vessels within the tumor while isotype LGNRs showed no such association (*Figure 8c–f, Figure 8—figure supplement 1*). The prevalence of anti-$\alpha_V\beta_3$ LGNRs around the edges of tumor capillaries is highly consistent with the expression pattern of $\alpha_V\beta_3$ in angiogenic vessels. Moreover, isotype LGNRs found outside of the vasculature were notably dispersed compared to extravascular anti-$\alpha_V\beta_3$ LGNRs, which often appeared in small clusters. While NPs are known to accumulate in tumors regardless of molecular specificity due to leaky vasculature, these results indicated that the enhanced extravascular accumulation of anti-$\alpha_V\beta_3$ LGNRs may have originated from specific binding of $\alpha_V\beta_3$ integrins present on the U87MG cells themselves.

## Discussion

The necessity of sample digestion with strong acids for ICP quantification effectively reduces an entire organ (a remarkably rich dataset by any measure) down to a single number representative of bulk NP accumulation. While the quantification offered by ICP is certainly valuable, it provides minimal insight into the patterns and mechanisms of NP uptake within individual cells or tissues. Unlike ICP methods, HSM-AD provides additional dimensions of anatomical detail at optical resolution to facilitate better understanding of the biology behind quantitative measurements of NP uptake.

The primary solution for dealing with the limitations of ICP has been to use EM, which provides excellent spatial resolution (at the nanometer scale) and particle sensitivity (down to individual nanoparticles). However, EM can only scan minimal fields of view—a typical transmission EM (TEM) image for studying NP uptake covers ~1 × 1 µm. For comparison, TEM scanning of the same area depicted in *Figure 3c* would require ~460,000 TEM images, which is infeasible for single tissue studies and virtually unrealistic for multiple-organ studies. The necessity of thin samples (~10 nm) for TEM imaging compared to samples analyzed using HSM-AD (~1 µm optical focus) would further multiply the number of TEM scans (>46 million) required for equivalent volumetric imaging. Other biodistribution techniques based on radioactivity (*Kreyling et al., 2015*; *Collingridge et al., 2003*), photoacoustic (*Poon et al., 2015*), and fluorescence (*He et al., 2010*) detection have been used previously as alternatives to ICP and TEM. By comparison, HSM-AD offers roughly 100-fold higher spatial resolution (~1 µm vs ~100 µm) than current fluorescence and photoacoustic biodistribution methods. Fluorescence-based methods may also suffer from high false positive detection arising from tissue autofluorescence, as has been observed for renal capsule tissue (*Poon et al., 2015*). While HSM-AD was excellently suited for exploring the sub-organ localization of NPs, it has been observed that radiolabeling approaches may be poorly-equipped for accurately determining particle distribution within

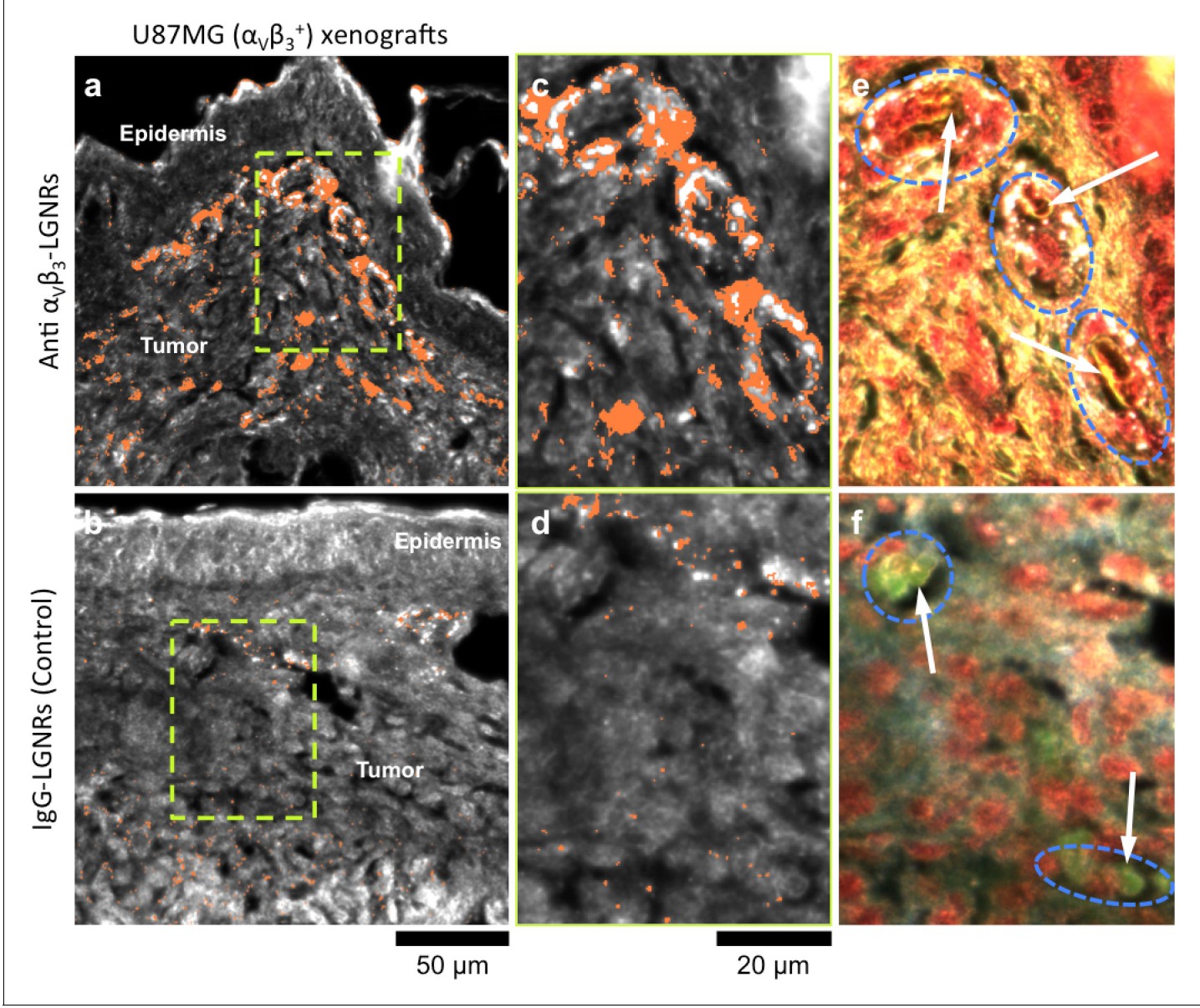

**Figure 8.** Active molecular functionalization affects nanoparticle uptake quantitatively and spatially within target tissues. (a,b) HSM-AD images of sub-dermal U87MG tumor xenografts from mice injected with LGNRs display distinct accumulation patterns depending on the molecular specificity of the LGNR surface coating. Anti-$\alpha_V\beta_3$ LGNRs exhibit 7.5-fold greater accumulation in tumor tissue (a) than spectrally-identical LGNRs with non-specific IgG antibody coating (b) (n = 4 FOVs for Anti-$\alpha_V\beta_3$ LGNRs, n = 5 FOVs for IgG-LGNRs, two-tailed Student's t-test, p=0.0041). The greater uptake of anti-$\alpha_V\beta_3$ LGNRs may result in part from specific LGNR binding to $\alpha_V\beta_3$ integrin, which is over-expressed by U87MG cells. (c–f) Validation of HSM-AD images with dark-field images of slightly higher spatial resolution further indicates that a substantial portion of anti-$\alpha_V\beta_3$ LGNRs are located along the edges of small capillaries within the tumor tissue (c,e) while no such association is observed for IgG-LGNRs (d,f). This observation is consistent with the nature of angiogenic tumor vasculature, which is also characterized by high expression levels of $\alpha_V\beta_3$ integrin in the vascular endothelium. Individual erythrocytes within angiogenic capillaries are denoted by white arrows, and capillary edges are approximately outlined by dashed blue ovals (e,f). Discrete regions of anti-$\alpha_V\beta_3$ LGNRs were also observed outside of the tumor vasculature, presumably due to either (1) specific binding to $\alpha_V\beta_3$ integrin expressed by U87MG cells and/or (2) non-specific accumulation via the enhanced permeability and retention (EPR) effect characteristic of tumors. The absence of IgG-LGNR extravascular accumulation suggests the former of these mechanisms as the predominant source of anti-$\alpha_V\beta_3$ LGNR uptake in tumor tissue.

The following source data and figure supplement are available for figure 8:

**Source data 1.** Data for tumor uptake of targeted and untargeted LGNRs.

**Figure supplement 1.** HSM-AD images of additional tumor tissue sections resected after targeted and untargeted LGNR injections.

organs (*Kreyling et al., 2015*). Moreover, single-particle detection sensitivity was not demonstrated by any of these alternatives to ICP and TEM.

Biodistribution methods based on imaging mass spectrometry were recently demonstrated to enable sub-organ quantification of carbon nanomaterials (*Chen et al., 2015*). This impressive approach can obtain images of full mouse tissue sections (cm in scale), but the limited spatial resolution (50 μm) precludes the study of NP uptake within individual cells. Because detection relies upon particle fragmentation and ionization, it is unclear whether imaging mass spectrometry can achieve single particle sensitivity—calculations based upon the reported data indicate that ~10 (*von Maltzahn et al., 2009*) particles per pixel are required for detection. However, the cited spatial resolution negates many of the potential advantages of single-particle sensitivity such as direct observation of NP uptake by cells through endocytosis or adhesion to the cell membrane. More generally, while mass spectrometry provides an approach to biodistribution studies of certain materials, its use for identifying gold NPs has been constrained to NPs smaller than 10 nm and typically requires the inclusion of 'mass barcode' molecules as capping agents (such as alkanethiols) on the NP surface (*Zhu et al., 2008*; *Harkness et al., 2010*). Incidentally, gold NPs have previously been demonstrated as assisting matrices to improve mass spectrometry detection of biomolecules (*Su and Tseng, 2007*; *Huang and Chang, 2007*). Notably, many metallic NPs are compositionally similar yet spectrally distinct (for example, gold nanospheres, nanorods, nanoshells, etc.), which may confound results in mass spectrometry-based analysis of samples containing more than one NP species. We have demonstrated that HSM-AD can successfully identify gold nanoparticles with different spectra, shape, size, and composition in tissues. Furthermore, Nanoshells and LGNRs were discernible from each other in particle mixtures. While not directly tested in this work, HSM-AD may thus be capable of distinguishing such NPs from each other in tissues, enabling biodistribution studies of multiplexed NPs. Thus, we expect that HSM-AD will extend the advantages of high sensitivity and resolution to the analysis of a variety of metallic NPs with unique spectral properties.

As reported in previous studies (*Chen, 2014*; *Okamoto et al., 2000*), we observed that the scattering spectrum from gold NPs is heavily influenced by the local refractive index (n). Spectral shifts of ~80 nm were evident between preparations of LGNRs in water (n = 1) relative to the same particles prepared in CytoSeal (n = 1.5) (*Figure 2—figure supplement 7*), and similar shifts were observed for Nanoshells and GNS@SiO$_2$. Thus, in order to generate reliable spectral cluster libraries, it is critical to train the adaptive algorithm using images prepared in similar fashion to the samples being studied. The orientation of anisotropic NPs within a sample has also been shown to influence the observed spectrum (*Biswas et al., 2012*). Because LGNRs are anisotropic, variation in particle orientation may affect detection sensitivity in our case, but this effect appears to be negligible in light of empirically measured sensitivity values. Because it relies on spectral identification of NPs, HSM-AD cannot detect NPs that have shifted their spectrum markedly, such as concentrated NP aggregation within cells. Such spectral hybridization is most prominent in Kupffer cells within the liver, which likely results in an artificially low measure of LGNR uptake in that organ. However, the absence of these bright aggregates in hyperspectral images of uninjected control tissues confirms their identity as LGNRs. Because of plasmon hybridization, LGNR aggregates produce shifted spectra that can resemble the scattering from H&E-stained tissue, which can impede automated detection. Thus, future efforts to detect such aggregates should rely on the analysis of unstained tissues.

We tested alternate machine learning approaches including support vector machine (SVM) and logistic regression for nanoparticle detection. We found that unsupervised k-means outperformed these other methods when trained on images of tissues containing nanoparticles (*Liba and Shaviv, 2014*). Further advantages of unsupervised k-means include its ease of use and no need to pre-label samples for analysis. While HSM-AD based on k-means clustering provides a robust general platform for nanoparticle detection, it is conceivable that certain studies may benefit from learning methods tailored to address specific applications.

HSM-AD imaging simultaneously achieves excellent sensitivity and specificity for detecting NPs in tissues with sub-cellular resolution. In addition to improved diagnostic capabilities, the automated and adaptive features of HSM-AD enable standardized high-throughput analysis previously absent from biomedical HSM studies (*Roth et al., 2015*). Unlike Spectral Angle Mapping (the current gold standard for HSM image analysis), HSM-AD does not require the manual steps typically needed to create target spectral libraries, define particle intensity and size thresholds, filter false positives from

libraries, and calibrate angular tolerance for accurate classification on an individual image basis (*Figure 2—figure supplement 15*) (*Roth et al., 2015*). Along with an ability to image millimeter-scale fields of view on reasonable timescales (<30 min) and simple sample preparation, these properties make HSM-AD a favorable alternative to existing methods for characterizing NP biodistribution. Beyond biodistribution, this work demonstrates that HSM-AD can be used for post-injection validation of NP localization in target tissues as a function of surface modifications. Because HSM-AD is non-destructive, samples can be further analyzed by a variety of conventional microscopy techniques including immunohistochemistry to provide additional molecular detail. Collectively, the results presented herein indicate that HSM-AD provides a new approach for studying interactions of cells and whole tissues with spectrally unique NPs commonly used in biomedical imaging and therapeutic studies.

## Materials and methods

### LGNR preparation

LGNRs were synthesized using methods adapted from Ye *et al* (*Ye et al., 2013*). LGNRs were characterized using Transmission Electron Microscopy (TEM), visible/near-infrared spectrometry, and dark-field hyperspectral microscopy. As-synthesized LGNRs were prepared for biological use by removing excess CTAB from solution and coating the particles with poly(sodium 4-styrenesulfonate) (PSS, MW 70 kDa) as previously reported (*SoRelle et al., 2015*). PSS-coated LGNRs were then conjugated with IgG isotype antibody (clone eB149/10H5, eBioscience) for use in sub-organ biodistribution experiments. We also prepared LGNRs targeted to $\alpha_V\beta_3$ integrin (a cell-surface receptor that is overexpressed in angiogenic vasculature within tumors) by conjugating LGNRs with anti-$\alpha_V\beta_3$ antibody (clone 23C6, eBioscience).

### Animal experiments and sample preparation

Healthy female nude (*Foxn1$^{nu/nu}$*) mice (6–8 weeks old, Charles River Labs) were anesthetized with 2% isoflurane by inhalation and intravenously injected with 250 μL of IgG isotype-coated LGNRs at optical density (OD) 470. In separate experiments, mice bearing U87MG tumors in the right ear pinna were injected with either IgG isotype-coated LGNRs or anti-$\alpha_V\beta_3$-coated LGNRs. Additional details of these experimental protocols can be found in the literature (*SoRelle et al., 2015*; *Liba et al., 2016*). In nanoshell experiments, nude mice were injected with 200 μL of OD 50 (2.5 mg/mL) nanoshells composed of 119 nm silica cores and 14 nm-thick gold shells with PEG coating (Nano Composix, San Diego, CA). In GNS@SiO$_2$ experiments, healthy female Balb/C mice were anesthetized as described previously and injected intravenously with 150 μL of 0.8 nM GNS@SiO$_2$ particles composed of 60 nm gold cores and 30 nm-thick SiO$_2$ shells (Oxonica, Mountain View, CA). For all experiments, mice were euthanized 24 hr (or 2 hr, for GNS@SiO$_2$) post-injection, and tissues including kidney, liver, lung, spleen, thigh muscle, and (when applicable) tumor were immediately resected and preserved in 10% formalin. Tissues were also resected from uninjected mice for control imaging experiments. These tissues were subsequently embedded in paraffin and sectioned into 5 μm thick samples. Sections were prepared with and without H&E stains and mounted on microscope slides using CytoSeal 60 (Electron Microscopy Sciences) as the mounting medium. All animal experiments were performed in compliance with IACUC guidelines and with the Stanford University Animal Studies Committee's Guidelines for the Care and Use of Research Animals. Experimental protocols (APLAC #s 27499 and 29179) were approved by Stanford University's Animal Studies Committee.

### Imaging system

All tissue samples were imaged with a modified dark-field microscopy setup as shown in *Figure 1—figure supplement 1.* Light from a broadband halogen lamp was coupled via an optical fiber into a custom dark-field condenser (CytoViva, Auburn, AL), which produced a light cone for sample illumination. Light scattered from the sample was collected using either a 40x magnification dark-field air objective lens (Olympus UPlanFLN 40x, 0.75 NA) or a 100x magnification oil immersion objective lens (Olympus UPlanFLN 100x, 1.3 NA) and directed to one of two cameras depending on detection mode. Conventional dark-field and hyperspectral images were collected for all samples in this study. Conventional dark-field images were collected with a Dagexcel-M cooled camera (Dage-MTI,

Michigan City, IN). Hyperspectral images were collected with a hyperspectral camera (iXon$_3$, Andor, Belfast, UK). Each image has 509 × 512 pixels. With a 40x lens, the sampling resolution is 410 × 408 nm, which produces a 209 × 209 µm field of view. With a 100x lens, the sampling resolution is 163 × 160 nm. Only *Figure 5* shows images acquired at 100x magnification. The spectrum from each pixel was acquired with 361 uniform samples at wavelengths ranging from 400 nm to 1000 nm. The acquired raw spectra of each pixel were lamp-normalized using the Cytoviva software package (ENVI 4.8) and exported after normalization.

## Data processing and automatic biodistribution detection

Processing of the hyperspectral images was done with Matlab (Mathworks, Natick, MA). Hyperspectral images were created by color-coding the spectrum by integrating over three bands. The band centers were 800.0 nm, 700.6 nm and 526.2 nm for the red, green and blue channels, respectively. The integration was weighted by a Gaussian window with a width of 80 spectrum samples. Each channel was scaled separately for optimal viewing.

Automatic detection of NPs required preprocessing the spectra prior to training and classification. First, due to noise at lower wavelengths, the spectra were truncated to disregard values below a cutoff of 566 nm. As part of HSM-AD, we initially segmented the image into background, tissue, or potential NPs based on each pixel's average intensity across its spectrum. This segmentation allowed a more accurate calculation of the biodistribution by measuring the number of pixels that correspond to tissue inside the field of view. The segmentation of potential NPs helped to avoid classifying low intensity edges that may be falsely detected as NPs due to chromatic aberrations. Before measuring the intensity of each pixel, we also applied a correction for vignetting. Vignetting is a common artifact in photography and microscopy in which image brightness is reduced at the periphery compared to the image center. We assumed a natural illumination fall-off that follows the 'cosine fourth' law, in which the light fall-off is proportional to the fourth power of the cosine of the angle at which the light impinges on the sensor. We measured the radial falloff of several images and found that it can be approximated by $cos^4(\theta)$, in which $\theta = tan^{-1}(R/d)$, where $R$ is the calculated distance from the center of the image and $d$ was found by fitting to be 2 mm. In order to correct the vignetting, we divided the intensities of each field of view by $cos^4(\theta)$. Next, we calculated the segmentation thresholds adaptively for each image (to account for different exposure times and variations in tissue scattering). The thresholds were obtained by analyzing the histogram of pixel intensities of each image (*Figure 1—figure supplement 2*). The histogram was calculated with 510 bins and then re-sampled (using interpolation) every 5 intensity units. Pixels with the lowest intensities were segmented as background. The threshold for segmenting the background was calculated as the first minimum of the histogram (minHist) multiplied by a user-defined parameter (which is slightly larger than 1) to allow fine tuning of the background threshold. Next, we assumed that pixels representing tissue without NPs have a relatively consistent intensity lower than that of NPs and therefore correspond to a peak in the histogram. The threshold for differentiating between pixels that correspond to tissue and those that can be potential NPs can be determined by the peak of the histogram (peakHist) multiplied by a pre-defined parameter (larger than 1) which allows tuning of the threshold. This intensity-based segmentation was confirmed to be effective by comparing results over 20 separate fields of view from all analyzed tissue types. Pixels segmented as potential NPs were preprocessed by smoothing their spectra using a Savitzky-Golay algorithm (*Orfanidis, 1995*) implemented by a Matlab's built-in function. Next, we normalized the spectrum of each pixel by the maximal intensity across its spectrum. Training and classification were performed only on pixels which were segmented as potential NPs.

Training of the k-means algorithm (*Bishop, 1995*) was initially done with 3, 4, and 5 clusters on 6 images of injected tissue sections, from which ~500,000 pixels were binned into the potential LGNR group. The 4 clusters found by the k-means algorithm matched the expected spectra of the injected and stained tissue. One of the spectra automatically matched the spectrum of LGNRs, owing to their distinct spectrum compared to tissue and staining dyes, two of the spectra represent the Hematoxylin and Eosin stains, and the fourth is an intermediate cluster representing the sum of both stains (see *Figure 1—figure supplement 4*). Testing the algorithm on uninjected tissue samples showed false detections of LGNRs near the edges of the tissue. We attributed these false positives to spectral red-shifting caused by chromatic aberrations due to the larger point spread function of longer

wavelengths compared to the smaller point spread function of shorter wavelengths. To minimize the number of these false positives, we manually added a cluster representing the spectrum of the chromatic aberrations by averaging the spectra of falsely detected pixels from uninjected samples. Indeed, this cluster showed a spectral peak near the resonance of the LGNRs, albeit much broader. The initial clusters found by k-means with the added cluster representing chromatic aberrations were used for classification of pixels as LGNR+ or LGNR- with a nearest centroid (or nearest neighbor) classifier, based on the Euclidean distance to the cluster centers. We then measured the sensitivity and specificity and also qualitatively assessed the results on injected tissue section using the algorithm with different numbers of clusters. We chose to use the cluster library which includes 5 clusters (4 obtained through k-means and the 5th added manually) because it produced the best results and clusters that were more consistent with the actual spectra present in the samples. Several other machine learning algorithms were also explored for this purpose, including support vector machine (SVM) and logistic regression, but k-means yielded better results (*Liba and Shaviv, 2014*). The classification results are presented as detection maps in which the average intensity at each pixel is displayed in grayscale and the pixels that are LGNR+ are shown in orange. Similar training and classification steps were performed for samples containing GNS@SiO$_2$ (3 clusters used) and Nanoshells (5 clusters for images of *ex vivo* tissues, 2 clusters for Nanoshells + LGNRs).

We characterized the sensitivity and specificity of HSM-AD by three methods. First, we measured the false positive and true negative rates in uninjected tissue samples to obtain the specificity. Note that false positives and true negative rates can be measured reliably from tissue-only samples due to the absence of LGNRs. Next, we measured the false negatives and true positives in an image of pure LGNRs in mounting media on a glass slide and obtained the detection sensitivity. Note that false negatives and true positives can be measured reliably from LGNR-only samples due to the absence of tissue and scattering media other than the particles themselves. In order to calculate the sensitivity and specificity for detecting LGNRs in tissue samples, we created a 'ground truth' data set by randomly choosing > 200 pixels, half of which were detected by the algorithm to contain LGNRs. We then manually (and in a manner blind to the results from the algorithmic classification) determined whether pixels were LGNR+ or LGNR- based purely on observing their raw spectra and looking for the unique plasmonic peak of LGNRs. Low-intensity pixels that were considered as the background or tissue were not considered in this calculation. By comparing the ground truth to the results of the automated algorithm we obtained the number of true positives, true negatives, false positives and false negatives, and used them to calculate additional measures of sensitivity and specificity. Confidence intervals for these measurements, presented in *Figure 2—figure supplement 1*, were calculated using the 'log method' (*Altman et al., 2000*) with the aid of a statistical calculator (*MedCalc Software, 2016*).

The biodistribution in each field of view was measured as the relative LGNR signal in an image. This measurement takes into account the amount of tissue versus background in a frame and also the signal of the detected LGNRs. For this calculation, we refer to LGNR signal as the average intensity around the plasmonic peak of the LGNRs (833–988 nm). The relative LGNR signal is the sum of the LGNR signal over LGNR+ pixels divided by the number of tissue pixels (i.e., the number of all pixels minus the number of background pixels) and divided by the median LGNR signal over the LGNR+ pixels in the field of view. For whole-organ analysis, we measured the relative LGNR signal in four fields of view for each organ (taken from the same mouse) and calculated the mean and standard deviation for each organ. For the sub-organ calculation, the measurement of relative LGNR signal yielded results with substantial variability due to a small number of pixels and high variability of intensity within several regions of interest. Therefore, a simpler pixel ratio (termed pixel coverage) was calculated by dividing the number of LGNR+ pixels by the number of tissue pixels for each region of interest (ROI). ROI maps are presented for each field of view in *Figure 4* as well as for the additional images used for quantification (*Figure 4—figure supplement 1*). The same calculations also apply for quantification of other particle types.

## Evidence of single particle detection

In order to determine whether HSM-AD is able to detect single LGNRs, we compared the theoretical point spread function of the microscope's 40x lens with the shape of the increased intensity caused by an isolated LGNR+ pixel. The diffraction-limited point spread function of the microscope can be approximated by a Gaussian with a standard deviation of 0.25 μm (based on a numerical aperture of

0.75 and wavelength of 910 nm) (*Lipson and Lipson, 2010*), which may increase in the case of defocusing. The cross section of intensity in the wavelengths around the plasmonic peak of the LGNRs (833–988 nm) showed a close resemblance to the theoretical spot size (*Figure 2—figure supplement 8*). This analysis supports the capability to detect single LGNRs in tissue samples.

## Acknowledgements

This work was funded in part by grants from the Claire Giannini Fund, the United States Air Force (FA9550-15-1-0007), the National Institutes of Health (NIH DP50D012179), the National Science Foundation (NSF 1438340), the Damon Runyon Cancer Research Center (DFS# 06-13), the Susan G Komen Breast Cancer Foundation (SAC15-00003), the Mary Kay Foundation (017-14), the Donald E and Delia B Baxter Foundation, the Skippy Frank Foundation, a seed grant from the Center for Cancer Nanotechnology Excellence and Translation (CCNE-T U54CA151459), and a Stanford Bio-X Interdisciplinary Initiative Seed Grant. AdlZ is a Pew-Stewart Scholar for Cancer Research supported by The Pew Charitable Trusts and The Alexander and Margaret Stewart Trust. EDS wishes to acknowledge funding from the Stanford Biophysics Program training grant (T32 GM-08294). OL is grateful for a Stanford Bowes Bio-X Graduate Fellowship. CLZ is supported by the National Cancer Institute of the National Institutes of Health under award numbers K22 CA160834 and R21 CA184608. JLC acknowledges funding from the Victorian government of Australia in the form of a Victorian Postdoctoral Research Fellowship. We would like to thank Debasish Sen, Byron Cheatham, Jamie Uertz, and Michelle Rincon for technical assistance. We wish to thank Dor Shaviv for help with testing alternative detection methods.

## Additional information

### Funding

| Funder | Grant reference number | Author |
| --- | --- | --- |
| Stanford University | Biophysics Program, T32 GM-08294 | Elliott D SoRelle |
| Stanford University | Bowes Bio-X Graduate Fellowship | Orly Liba |
| Victorian Government of Australia | Postdoctoral Research Fellowship | Cristina L Zavaleta |
| National Cancer Institute | K22 CA160834 | Cristina L Zavaleta |
| National Cancer Institute | R21 CA184608 | Adam de la Zerda |
| Claire Giannini Fund | | Adam de la Zerda |
| U.S. Air Force | FA9550-15-1-0007 | Adam de la Zerda |
| National Institutes of Health | NIH DP50D012179 | Adam de la Zerda |
| Damon Runyon Cancer Research Foundation | DFS# 06-13 | Adam de la Zerda |
| Susan G. Komen | Breast Cancer Foundation, SAC15-00003 | Adam de la Zerda |
| Mary Kay Foundation | 017-14 | Adam de la Zerda |
| Donald E. and Delia B. Baxter Foundation | | Adam de la Zerda |
| Skippy Frank Foundation | | Adam de la Zerda |
| Center for Cancer Nanotechnology Excellence and Translation | CCNE-T U54CA151459 | Adam de la Zerda |
| Stanford Bio-X Interdisciplinary Initiative | Seed Grant | Adam de la Zerda |
| National Science Foundation | NSF 1438340 | Adam de la Zerda |

The funders had no role in study design, data collection and interpretation, or the decision to submit the work for publication.

## Author contributions

EDS, OL, Designed the study, Performed all experiments, Acquisition of data, Analyzed the data, Wrote the paper, Approved the final version of this work; JLC, CLZ, Performed experiments, Acquisition of data, Analysis and interpretation of data, Approved the final version of this work; RD, Prepared all histological sections, Conception and design, Acquisition of data, Approved the final version of this work; AdlZ, Designed the study, Analysis and interpretation of data, Wrote the paper, Approved the final version of this work

## Author ORCIDs

Elliott D SoRelle, http://orcid.org/0000-0002-3362-1028
Orly Liba, http://orcid.org/0000-0003-4625-9838

## Ethics

Animal experimentation: All animal experiments in this study were performed in compliance with IACUC guidelines and with the Stanford University Animal Studies Committee's Guidelines for the Care and Use of Research Animals (APLAC Protocol #27499 and #29179).

## Additional files

### Supplementary files

• Source code 1. Contains all MATLAB code used for HSM-AD.

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
