## [Decision Letter]

[Editors’ note: this article was originally rejected after discussions between the reviewers, but the authors were invited to resubmit after an appeal against the decision.]

Thank you for submitting your work entitled "A hyperspectral method to assay the microphysiological fates of nanomaterials with single-particle sensitivity" for consideration by *eLife*. Your article has been reviewed by three peer reviewers, and the evaluation has been overseen by Gaudenz Danuser as the Reviewing Editor and Sean Morrison as the Senior Editor.

Our decision has been reached after consultation between the reviewers. Based on these discussions and the individual reviews below, we regret to inform you that your work will not be considered further for publication in *eLife*.

As you will see from the full reviews below, all reviewers praise the quality of the study and its relevance and timeliness. However, all three reviewers raised from different angles the same concern: How new and versatile is the technique. Reviewer #1 provides a number of papers that your approach should be carefully compared to – potentially with additional validation experiments. Similarly, Reviewer #3, an expert in machine learning, is not convinced that the approaches taken are all state-of-the-art. Some cross-validation with other methods would help easing this concern for a future reader of a publication. Reviewer #2 raises the most critical point: although the claim of generality is made, the data presented relies on a single type of nano-particles. In the discussion among the reviewers following the initial evaluation it became clear that manuscript could only be considered if data from multiple nano-particles were shown. We suppose that acquisition of such complementary data will exceed the typical time *eLife* grants for a revision. Therefore, we have decided at this point to reject the manuscript. That said, there is significant merit in the combination of approaches, leading to a significant result. Thus, with a more thorough comparison of your approaches to others and a demonstration of applicability to other nano-particles we would consider a new submission of this work and make our best effort to send the manuscript back to the same editor and reviewers.

*Reviewer #1:*

In their work the authors propose a technique for quantifying nanoparticles distribution in histological samples. Hyperspectral imaging is based on the collection of images containing spectral information across a large (relatively speaking) range of the electromagnetic spectrum. Typical applications are found for military, geoscience, or environmentally studies. Here for each pixel within an image for example the signal across the visible or infrared spectrum is collected and hyperspectral data cubes are built and analyzed. Because different objects possess different optical properties, knowing *a priori* specific signatures allow classifying the information present within a scene.

In recent years this technique has found quite some success within the biomedical imaging field. Because metallic nanoparticles scatter light quite strongly at specific "resonant" wavelengths, the combination of darkfield microscopy and an hyperspectral approach has made it possible to successfully detect nanoparticles at high imaging speed and to separate them from the cellular background. Also the technique offers greater resolution when compared to other tools for studying biodistributions.

In the manuscript the authors report specifically on an imaging processing method to quantify nanoparticles distributions in tissue samples. The imaging setup is a standard one from CytoViva for "enhanced darkfield microscopy" equipped with a CCD camera for hyperspectral imaging. Previous work in the field is already present in my opinion. See for example a review paper from Brenner's group detailing recent works ("Hyperspectral microscopy as an analytical tool for nanomaterials"). Different groups have also presented several works demonstrating single-nanoparticle detection. See for example "Single-nanoparticle detection and spectroscopy in cells using a hyperspectral darkfield imaging technique" and others from Musken's group, and also another nice paper from Meunier group "Hyperspectral darkfield microscopy of PEGylated gold nanoparticles targeting CD44-expressing cancer cells" where 3D nanoparticles tracking is also demonstrated.

Because the authors concentrate on *ex vivo* tissue sections I found strange that very recent intriguing work from Brenner's group is not cited considering it is focusing basically on a very similar subject (i.e. "Identification of Metal Oxide Nanoparticles in Histological Samples by Enhanced Darkfield Microscopy and Hyperspectral Mapping" on JoVE 2015). It would have been nice to see a discussion and analysis of this work and in which respect the presented work differs from the cited one.

Having said that, I found the paper very interesting and very well written. There is a lot of work and data are very compelling. Also I think it could be of great interest. My only concern deals with the novelty of it (specifically see the paper mentioned above from *JoVE*).

Also, because the paper deals with the development of a machine learning algorithm I found myself a little bit in difficulty giving a judgement in this regard because I'm not a specialist of this particular field and I'm familiar with only the most common and basic approaches. From what I've seen a lot of sophisticated work has been done for hyperspectral classification in areas outside from the biological one (e.g. recently deep-learning based approaches for feature extraction for hyperspectral imaging have been implemented from different groups). Therefore, I honestly cannot judge in this regard and perhaps maybe someone with a specific imaging processing background in the field of hyperspectral classification (not necessarily limited to biomedical microscopy, but a more broad one) could perhaps provide more insights into it and comment on the novelty of the computational approach used here.

*Reviewer #2:*

Overall, this is an interesting paper that seems to have been executed with significant care. The figures are of high quality and the text is well written.

Strengths:

The authors use hyperspectral imaging to detect large gold nanorods (100 x 30 nm) in *ex vivo* tissues with single particle detection capabilities and single micron resolution.

They analyze histologically stained tissue slices from various organs, and the appropriate controls. From a technical standpoint, everything looks fine. I was expecting this technique to not work for aggregated NPs, especially in liver Kupfer cells, and they report exactly that, the honesty of which I appreciate.

Unlike many papers that provide descriptive multivariate models for their data, in this one, they use a training dataset to build the model, and then they use the model on unknown samples. This shows that the model has a high predictive power. I would be curious to see how well this method could distinguish between different types of plasmonic particles. Also, the particles used are ~30x100 nm2.

Weaknesses:

In my opinion the biggest weakness of this paper is that the technique is only demonstrated for one single nanoparticle type, and especially one that is not commonly used, i.e. large gold nanorods that are 100 x 30 nm in size.

However, in the Discussion the authors specifically claim that their technique is applicable to many nanoparticle shapes "(for example, gold nanospheres, nanorods, nanocages, etc.)" […]. "ABIDE is capable of distinguishing such NPs from each other by spectral differences, enabling biodistribution studies of multiplexed NPs." This is not only a bold claim, which importantly is not supported by any data.

The technique is only interesting enough for a broad audience if multiple different nanoparticle shapes such as spheres, cages, regular sized nanorods etc., can be analyzed with this method.

And my concern is that the technique will run into issues with discriminating some of the other shapes from cellular components, as can already be predicted when considering the curves in "Figure 1—figure supplement 4". The large gold nanorods have a plasmonic peak in the near infrared, which may be much easier to discriminate by the machine-learning algorithm than for shapes such as e.g. spheres.

The authors need to show convincingly that this technique works for the other nanoparticle shapes as they claim.

*Reviewer #3:*

The paper presents methods and results of combining mathematical modeling, data analysis, hyper spectral imaging, related to imaging of nano materials with applications in cancer, angiogenesis, and others. I find this area really interesting, and the potential impact large, although tissue and cell physiology are not my area of expertise. Though I find the ideas very stimulating, the main issue that I have is that the mathematical modeling/data analysis used here is not very clear, and may not be near top shelf work. Combined with certain choices of validation expanded below, my understanding is that the contributions to imaging methodology is not sufficiently novel, nor inspire a lot of confidence that results are as best as possible. See below for detailed comments.

Regarding modeling and data analysis, a few questions. The topic of discerning the contributions of different elements (in this case nano materials versus others such as Eosin, Hematoxylin, etc.) from spectral measurements is a well studied one. While I understand the intricacies of this imaging experiment are not the same as other more well studied spectral unmoving problems, I'm not convinced that the wealth of other methods for linear and nonlinear unmixing don't apply here. In the paper I did not find any discussion related to this. Classification methods of the type authors claim to have attempted (nearest neighbor, SVMs, etc.) could be applied on such unmixed data, and to me this would constitute a more standard way of doing things. As is, the methodology regarding data analysis coupled to imaging does not seem very novel, and it is not presented in a way that can be related to many other already proposed methods to other seemingly similar applications.

Regarding the data analysis for validation, a couple of things seemed unclear to me. For the detection of false positives and false negatives, were these evaluations performed separately? And if so, is the validation criterion computed by checking whether there is one single pixel indicating nonmaterial (in case of false positives) present? Similar comments for the reverse situation (false negatives). It would be necessary to know the specifics of these details better, and even better have the data (e.g. histograms and if pixel counts are used, how thresholds are utilized). Also, how is the user defined parameter (it seems a manually selected threshold is used to initially determine if a pixel is potentially LGNR) handled during validation? Is data used in the validation stage also used by the user when selecting this parameter? It is a little confusing when in the subsection “Data processing and automatic biodistribution detection” the authors comment "We then manually (and in a manner blind to algorithmic classification)…". How is this possible given that for a positive detection the pixel must be classified as potential LGNR first? Am I missing something, or is it possible that authors are mixing training and testing data?

A few more comments below:

The Results section reads more like a Methods section. If format is to be followed, I'd suggest only describing results in the Results section.

Results, second paragraph: K-means is most commonly referred as a clustering algorithm. Once clusters are identified, one can use this information to design a multitude of classification methods, but authors must specify which they used. Presumably the simplest would be the nearest neighbor cluster center. Are the cluster centers utilized as ground truth?

[Editors’ note: what now follows is the decision letter after the authors submitted for further consideration.]

Thank you for resubmitting your work entitled "A hyperspectral method to assay the microphysiological fates of nanomaterials with single-particle sensitivity" for further consideration at *eLife*. Overall, your revised article has been favorably evaluated by Sean Morrison (Senior editor), a Reviewing editor, and three reviewers.

The manuscript has been improved but there are some remaining issues that need to be addressed before acceptance. They mainly concern the tendency of 'over-selling' some of the approaches and accomplishments of the work. We especially encourage you to reconsider the use of an acronym for this technology. We are concerned about the use of acronyms as cheap eye-catchers. Moreover, a new expert in machine learning has been brought to the panel because previous Reviewer #3 was unavailable. Given the rapid pace of the machine learning field it seems inappropriate to call the application of a clustering method (and even a very classic one, which is taught in undergraduate computer science course) 'machine learning'. It is perfectly fine to use standard methods if they solve the task, but there is a discomfort among reviewers that machine learning is merely used as a catchy phrase in this case. This could fire back on your work. We thus encourage you to revise some of the language in your manuscript, including the Abstract, and to address the few other comments listed below.

*Reviewer #1:*

I've found that the authors’ comments are very appropriate and to the point even though in some cases I'm not too familiar with some of the points discussed. My major concern is always related to my original comment. Previous similar work from Brenner using a basically identical setup hardware is already present in the literature. Because the scientific contribution of the submitted paper consists in proposing an analysis procedure based on a machine learning algorithm with the intent of extending the work of Brenner and co-workers and others, it is critical to determine the novelty of the proposed algorithm. Overall I found the data provided by the authors very compelling and particularly interesting.

*Reviewer #2:*

The authors are now showing feasibility data from two other nanoparticle shapes, and have therefore satisfactorily addressed my previous main concern whether ABIDE may be versatile enough with regards to different nanoparticle shapes and sizes.

There are a few remaining issues.

General comment: The revised version does not include any tracked changes or other markings to indicate where changes were made, which made the review of the paper quite difficult. I am making this comment not because I want to review another version with track changes, but to make it clear that this limitation may have reduced my ability to catch all remaining or new issues.

Specific comments:

1) In the PowerPoint slides the authors provided in their rebuttal (for the reviewers only), it says "Example of new in vivo data". I find this misleading, as I could not find any data in the entire paper that was acquired "in vivo". A reviewer who does not carefully examine the manuscript may be misled by the rebuttal summary slides that this is in fact all acquired in vivo and not catch this discrepancy between the summary and the actual paper. I am not expecting the authors to provide true in vivo data, but would like to clarify what the authors meant by that.

2) The title of the paper is overstated with regards to claiming "single particle sensitivity" and this needs to be changed or else would be misleading. Figure 2—figure supplement 8 is the only figure that shows any data that would support that, and only in the very large gold nanorods (which by some definitions would not represent a nanoparticle). "Likely" as is stated in the figure legend is probably an honest assessment by the authors, but not enough to make such a major claim, and there is no evidence for this to work in the other two nanoparticles that are now included. I suggest replacing "…with single particle sensitivity" with "… in tissue sections" or "… in histological slices".

I would be willing to accept the manuscript pending these clarifications/changes if the other reviewers agree that their areas of expertise were addressed sufficiently as well.

*Reviewer #4:*

This revised manuscript describes an interesting and straightforward development of an approach for detecting nanoparticles in hyperspectral dark field images. It has been significantly improved based on comments in the initial reviews. The results presented demonstrate an impressive ability to detect and quantify these nanoparticles in tissue images. From an image processing/analysis/machine learning point of view, the approach is not novel or instructive (clustering spectra, especially with manual tuning, barely qualifies to be called machine learning, and does not seem to warrant a new acronym). Hence the significance of the manuscript must derive from the future importance of the method's application, something this reviewer is not qualified to judge.

---

## [Author Response]

[Editors’ note: the author responses to the first round of peer review follow.]

*As you will see from the full reviews below, all reviewers praise the quality of the study and its relevance and timeliness. However, all three reviewers raised from different angles the same concern: How new and versatile is the technique. Reviewer #1 provides a number of papers that your approach should be carefully compared to – potentially with additional validation experiments. Similarly, Reviewer #3, an expert in machine learning, is not convinced that the approaches taken are all state-of-the-art. Some cross-validation with other methods would help easing this concern for a future reader of a publication. Reviewer #2 raises the most critical point: although the claim of generality is made, the data presented relies on a single type of nano-particles. In the discussion among the reviewers following the initial evaluation it became clear that manuscript could only be considered if data from multiple nano-particles were shown. We suppose that acquisition of such complementary data will exceed the typical time eLife grants for a revision. Therefore, we have decided at this point to reject the manuscript. That said, there is significant merit in the combination of approaches, leading to a significant result. Thus, with a more thorough comparison of your approaches to others and a demonstration of applicability to other nano-particles we would consider a new submission of this work and make our best effort to send the manuscript back to the same editor and reviewers.*

We thank the editors and reviewers for their fair and constructive feedback on our manuscript. Below, we have provided detailed responses to the concerns raised by each reviewer. In summary, the following key revisions have been made to our manuscript:

a) New Nanoparticles Tested with Favorable Results:Extensive ex vivo data are now presented for two additional nanoparticle – Gold Nanoshells and silica-coated Gold Nanospheres (GNS@SiO_2_) – each of which exhibit distinct spectral properties, shapes, and compositions from the LGNRs we reported in the initial manuscript (please see Figure 6 and Figure 7, as well as all related figure supplements). We also demonstrated the ability to spectrally multiplex Nanoshells *and* LGNRs using our method, despite their highly overlapping spectrum (see Figure 6—figure supplement 3). Specific details of each new particle type and experimental result are noted in the response to Reviewer #2’s comments.

b) Direct Comparison of ABIDE with Existing Hyperspectral Methods: We have updated the Introduction and Discussion sections of our manuscript with comparisons to the existing studies recommended by Reviewer #1. Extensive comparisons are provided in the response to Reviewer #1’s comments below. An additional experimental comparison with the cited spectral angle mapping (SAM) approach is also now included in the manuscript (Figure 2—figure supplement 15).

c) Cross-Validation and Comparison with Other Machine Learning Approaches: The results of k-means analysis were compared with those from other prevalent machine learning methods including SVM. A detailed report of results from these various methods is listed as Liba and Shaviv, 2014 in our manuscript, and our rationale for choosing k-means analysis is summarized within the response to Reviewer #3. Additional discussion of alternate machine learning approaches has been added to the manuscript.

Summary of new data:

Figure 2—figure supplement 1: now contains raw data for sensitivity/specificity (panel b);

Figure 2—figure supplement 15: new data comparing ABIDE to spectral angle mapping (SAM);

Figure 6 and all supplements: new data for ABIDE detection and spectral unmixing of gold nanoshells;

Figure 7 and all supplements: new data for ABIDE detection of gold nanospheres.

*Reviewer #1:*

*In their work the authors propose a technique for quantifying nanoparticles distribution in histological samples. Hyperspectral imaging is based on the collection of images containing spectral information across a large (relatively speaking) range of the electromagnetic spectrum. Typical applications are found for military, geoscience, or environmentally studies. Here for each pixel within an image for example the signal across the visible or infrared spectrum is collected and hyperspectral data cubes are built and analyzed. Because different objects possess different optical properties, knowing a priori specific signatures allow classifying the information present within a scene.*

*In recent years this technique has found quite some success within the biomedical imaging field. Because metallic nanoparticles scatter light quite strongly at specific "resonant" wavelengths, the combination of darkfield microscopy and an hyperspectral approach has made it possible to successfully detect nanoparticles at high imaging speed and to separate them from the cellular background. Also the technique offers greater resolution when compared to other tools for studying biodistributions.*

*In the manuscript the authors report specifically on an imaging processing method to quantify nanoparticles distributions in tissue samples. The imaging setup is a standard one from CytoViva for "enhanced darkfield microscopy" equipped with a CCD camera for hyperspectral imaging.*

We thank the reviewer for this concise overview of hyperspectral imaging and its applications for context. Similar descriptions can be found in the Introduction and Discussion sections of our manuscript.

*Previous work in the field is already present in my opinion. See for example a review paper from Brenner's group detailing recent works ("Hyperspectral microscopy as an analytical tool for nanomaterials"). Different groups have also presented several works demonstrating single-nanoparticle detection. See for example "Single-nanoparticle detection and spectroscopy in cells using a hyperspectral darkfield imaging technique" and others from Musken's group, and also another nice paper from Meunier group "Hyperspectral darkfield microscopy of PEGylated gold nanoparticles targeting CD44-expressing cancer cells" where 3D nanoparticles tracking is also demonstrated.*

We thank Reviewer #1 for bringing these references to our attention. We have added a comparison of these previous methods with the method described in our manuscript to the Discussion section. In general these existing works should be acknowledged (they are now cited in our revised manuscript), but we do not believe they compromise the novelty or utility of our current study. Specifically, our manuscript reports new capabilities for diagnostic evaluation of particle identification in HSM images, improvements for standardizing HSM analysis and reproducibility, validated single-particle detection in animal tissues, the first demonstration of HSM for systemic biodistribution studies, and the novel nanoparticle microbiodistribution data itself (now 3 particle types). Further details are provided below:

“Hyperspectral microscopy as an analytical tool for nanomaterials” (Brenner): This review article is listed in our original manuscript. We cited this review (and several other studies the review cites) in the Introduction section of our manuscript, as it provides very useful context for biological applications of hyperspectral imaging. Importantly, the review by Brenner also highlights the outstanding problems and limitations of hyperspectral imaging, several of which can be resolved by our method (summarized below). Please see the following limitations mentioned in the Brenner review article and how our methods can address these limits:

“More significantly is that while [Hyperspectral Imaging] enables the detection of nanomaterials, no study has precisely quantified the rate of false-positive identification or assessed its efficacy in the presence of stains used in histology or immunohistochemistry.” (Brenner, WIRE Nanomed Nanobiotechnol 2015)

We provide false-positive quantification measurements validated through several approaches, along with full diagnostic characterizations of detection sensitivity and specificity. Notably, these capabilities are demonstrated in stained histological samples. Sentences highlighting this existing problem have been added to the Introduction and Discussion.

“[Hyperspectral Imaging] is also sufficiently novel that there are few standardized methods to be applied, with the result that each lab may use differing source power, exposure time, or other parameters which can result in significant variance in spectra of similar materials analyzed under different conditions.” (Brenner, WIRE Nanomed Nanobiotechnol 2015)

Because of the adaptive nature of our algorithms with respect to a given image’s intensity histogram, the detrimental effects of variance in incident illumination power and exposure time can be minimized using our method. This provides potential benefits in terms of result reproducibility and the development of standardized methods, which is now emphasized in the manuscript.

"Single-nanoparticle detection and spectroscopy in cells using a hyperspectral darkfield imaging technique" (Muskens): Reviewer #1 is correct that this paper provides a nice demonstration of single particle detection with hyperspectral imaging – we have now added the paper by Muskens as a useful reference for validation of our own single particle detection capabilities (i.e., validation of empirical signals against theoretical optics predictions, etc.). We do wish to clarify that the cited Muskens paper demonstrates single particle detection in pure nanoparticle samples and, in the case of another Muskens paper (Fairbairn et al., Phys. Chem. Chem. Phys. 15(12) 2013), in cell culture. It should be noted that the ability to detect particles in cell culture does not guarantee that such detection will be possible in tissues (especially stained sections) due to the complex spectral scattering environment. By comparison, we use our method to demonstrate single-particle detection capabilities in such tissue sections. We see this as a distinction of our manuscript relative to existing work.

“Hyperspectral darkfield microscopy of PEGylated gold nanoparticles targeting CD44-expressing cancer cells” (Meunier): As Reviewer #1 notes, this paper also demonstrates single particle detection. As for the papers from Muskens’ group, we note that this work is limited to particle detection in cell culture rather than whole tissues following systemic nanoparticle administration. For completeness, we have now incorporated this reference into our manuscript’s Introduction section. We would like to clarify that the 3D particle tracking demonstrated in the paper from Meunier is achieved through image acquisition at multiple focal planes (z-axial scanning) in fixed samples followed by reconstruction – in principle, this approach could also be applied for 3D characterization of tissue sections using our methods.

A general note on the hyperspectral methods used by Muskens and Meunier: both groups implement a super-continuum tunable light source. While we expect that our methods can translate to hyperspectral imaging with laser-based sources, the success of spectral cluster identification may be influenced by the chosen spectral resolution (laser sampling density/sparsity). While not necessarily prohibitive, higher spectral sampling density with a tunable laser typically leads to longer scan times. Sufficiently high spectral resolution is important for the accurate classification of spectra into clusters. Also, as Meunier notes, the use of laser-based sources can introduce speckle that can corrupt image formation and quantification. Meunier also points out the following with respect to using laser-based hyperspectral imaging:

“It should be mentioned that additional experimental efforts are needed to obtain spectrally and spatially homogeneous imaging field comparable to the conventional white light illumination used in push-broom hyperspectral imaging systems (CytoViva, PARISS). Accurate spectral and spatial characterization of single and aggregated AuNPs is essential for the improvement of nanoplasmonic-based imaging, disease detection and treatment in complex biological environment” (Meunier, J. Biophotonics 8(1-2) 2015).

Thus, spectral stability and uniformity are further advantages of the system and methods we demonstrate relative to existing work in the field, although this is a technical point related more to optical setup and hardware rather than analytical methods.

*Because the authors concentrate on ex vivo tissue sections I found strange that very recent intriguing work from Brenner's group is not cited considering it is focusing basically on a very similar subject (i.e. "Identification of Metal Oxide Nanoparticles in Histological Samples by Enhanced Darkfield Microscopy and Hyperspectral Mapping" on JoVE 2015). It would have been nice to see a discussion and analysis of this work and in which respect the presented work differs from the cited one.*

Thank you to Reviewer #1 for bringing this work to our attention. Reviewer #1 is correct that the focus of the work by Brenner is related to our own study, and for this reason we have now added it as a reference (Roth et al., 2015) and addressed its relation to our work in the Discussion section as requested. For completeness, there are several advantages to our approach, which are described below:

Our method’s accuracy and practicality enable new applications of hyperspectral imaging: Brenner et al.describe (in excellent detail) the use of commercially-available ENVI software for analyzing hyperspectral images acquired using a CytoViva microscope (similar to the one we use in our study). As noted in the “Protocol” portion of the paper in JoVE, this analysis involves separate user-initiated steps for 1) appropriate selection of reference spectra for particles of interest, 2) removal of false-positive spectra, and 3) use of the reference spectra produced in 1) and 2) for spectral angle mapper (SAM) classification of images of interest. Within the SAM analysis steps themselves, users must manually set factors including intensity thresholds and target size parameters in order to obtain accurate (i.e., high-specificity, high-sensitivity) representations of the presence of a given nanomaterial in the sample. Following SAM classification, the authors note that mapped hyperspectral images are then analyzed by additional third party software (ImageJ) to obtain quantitative data on particle uptake/presence. Finally, the authors note that this quantitative data should be exported to other programs for statistical analysis. This process can then be repeated for additional images.

By comparison, the methods described in our manuscript achieve spectral identification, diagnostic validation, and quantification rapidly using an open-source code to process many images in a single run. Because the processing is automatic, we believe our method provides a means for higher throughput analysis, the importance of which should not be overlooked if the technique is to be used for biodistribution studies that require analysis of numerous images of various tissues.

Because it is adaptive, our method requires minimal user-defined inputs. In addition to manual definition of intensity thresholds (see below) or expected particle size (which may be highly variable depending on routes of uptake/accumulation in tissues and therefore cannot be assumed a priori), SAM classification requires users to define angular tolerance values that can heavily impact the resulting detection sensitivity and specificity (see Figure 2—figure supplement 15). This may introduce significant detection error that can be difficult to quantify.

The existing method reported by Brenner in JoVE produces spectral libraries through methods that are susceptible to manual biases and non-comprehensive sampling. Thus, libraries may not be faithfully reproduced by different users or for different samples. From the JoVE Protocol section “Creation of Reference Spectral Libraries”:

3.1.3 “…click onto pixels of interest on the datacube, particularly the brightest ones or those that can be confidently identified as representing the material of interest…take note particularly of their lowest and highest value and which wavelength corresponds to it.” The authors then note in section 3.1.4 that these user-observed values should be entered as parameters into the software’s “Particle Filter” function to identify pixels with spectra that will be added to the spectral library file used for subsequent image classification.

For images with more than 200,000 pixels, manual assessment of this type is likely inadequate to determine the relevant intensity and spectral ranges that will allow complete and accurate particle identification in subsequent steps. In other words, the results for any image analyzed with a spectral library defined in this way will only be as specific/sensitive as the initial user’s discretion allows. Accuracy and reproducibility among users and images (even for similar samples under different illumination) become significant concerns for this reason. Brenner’s group notes this specific issue in their review article from the same year as well.

Comparison of article scope and novelty: The cited work by Brenner successfully demonstrates the ability to identify nanomaterials of different composition following administration to samples of porcine skin. While this demonstration is relevant to our manuscript, we note the following novel aspects of our manuscript that are not within the scope of this or other previous hyperspectral imaging work:

The machine learning approach we implemented realizes several new improvements to existing issues in the field of hyperspectral imaging (as noted in the review by Brenner, see above). While the machine learning method is not novel per se, its application to improve the quality of biological hyperspectral image analysis is the first such demonstration to our knowledge.

While Brenner and several other groups referenced by our work have rightfully discussed the potential of hyperspectral imaging for performing studies of biodistribution, our manuscript provides the first empirical demonstration of hyperspectral imaging as a viable biodistribution technique. Thus in terms of potential clinical impact and future use, we believe our manuscript marks a significant advance for hyperspectral imaging as a biomedical resource.

In addition to demonstrating biodistribution capabilities, we provide novel data on the sub-organ localization of nanoparticles (LGNRs, Nanoshells, and GNS@SiO_2_) that have not been previously reported. The microbiodistribution data is itself a novel resource that can inform future uses (clinical or otherwise) of the studied nanoparticles.

We also report the first hyperspectral study of molecularly-targeted particle uptake in tissue (tumor xenografts from live animal models). [We note the previous demonstration of CD 44-targeted particle uptake in cell culture by Meunier et al.] While this data is itself unique, the greater takeaway is that ABIDE may be an ideal tool for future assessments of nanoparticles used for targeted imaging and therapy in animal models of disease.

The study by Brenner’s group in JoVE reports the detection of nanomaterials in tissue, but the authors do not assess single particle detection sensitivity in these tissues. Our manuscript provides one such demonstration of single-particle sensitivity in tissue.

The cited work does not demonstrate spectral unmixing capabilities to distinguish different particles types present in a single sample, which we have now demonstrated (see Figure 6—figure supplement 3).

*Having said that, I found the paper very interesting and very well written. There is a lot of work and data are very compelling. Also I think it could be of great interest. My only concern deals with the novelty of it (specifically see the paper mentioned above from JoVE).*

We thank Reviewer #1 for this very positive consideration of our work. We believe our work provides several notable novel advances relative to current work in the fields of biodistribution and hyperspectral imaging. For more details on these novel aspects, please refer to previous addresses of existing literature.

*Also, because the paper deals with the development of a machine learning algorithm I found myself a little bit in difficulty giving a judgement in this regard because I'm not a specialist of this particular field and I'm familiar with only the most common and basic approaches. From what I've seen a lot of sophisticated work has been done for hyperspectral classification in areas outside from the biological one (e.g. recently deep-learning based approaches for feature extraction for hyperspectral imaging have been implemented from different groups). Therefore, I honestly cannot judge in this regard and perhaps maybe someone with a specific imaging processing background in the field of hyperspectral classification (not necessarily limited to biomedical microscopy, but a more broad one) could perhaps provide more insights into it and comment on the novelty of the computational approach used here.*

We thank Reviewer #1 for the positive review of our work and for the suggestion of several useful references that we have added to our manuscript. A more detailed discussion of the machine learning methods tested for our study is addressed in the response to Reviewer #3’s comments.

*Reviewer #2:*

*Overall, this is an interesting paper that seems to have been executed with significant care. The figures are of high quality and the text is well written.*

*Strengths:*

*The authors use hyperspectral imaging to detect large gold nanorods (100 x 30 nm) in ex vivo tissues with single particle detection capabilities and single micron resolution.*

*They analyze histologically stained tissue slices from various organs, and the appropriate controls. From a technical standpoint, everything looks fine. I was expecting this technique to not work for aggregated NPs, especially in liver Kupfer cells, and they report exactly that, the honesty of which I appreciate.*

We thank the reviewer. Anecdotally, Reviewer #2 may find it interesting that ABIDE detection of silica-coated gold nanoshells (GNS@SiO_2_) in Kupffer cells did not seem to be impeded by spectral hybridization. We suspect this may be due in part to the steric effects of the silica shell around the gold core, however it may also be attributed to smaller spectral shifts relative to the original peak upon aggregation. (See liver images in Figure 7 and Figure 7—figure supplement 5).

*Unlike many papers that provide descriptive multivariate models for their data, in this one, they use a training dataset to build the model, and then they use the model on unknown samples. This shows that the model has a high predictive power.*

Thank you to Reviewer #2 – we appreciate these compliments on our work.

*I would be curious to see how well this method could distinguish between different types of plasmonic particles. Also, the particles used are ~30x100 nm2.*

As part of the revisions to our work, we have included new data detailing the detection of two additional plasmonic particles: gold nanoshells (Nanoshells) and silica-coated gold nanospheres (GNS@SiO_2_). These particles exhibit very different spectra from the LGNRs originally tested. In particular, the GNS@SiO_2_ have ~550nm SPR vs ~850nm SPR for LGNRs. The size, shape, and material composition/surface coating of both Nanoshells and GNS@SiO_2_ are also distinct from LGNRs. Results for these two new particle types are found in Figure 6 and 7. Please also refer to new results for spectral unmixing of gold Nanoshells and LGNRs (Figure 6—figure supplement 3). We hope this (and other) new data helps answer Reviewer #2’s questions regarding the ability to distinguish between plasmonic particles.

*Weaknesses:*

*In my opinion the biggest weakness of this paper is that the technique is only demonstrated for one single nanoparticle type, and especially one that is not commonly used, i.e. large gold nanorods that are 100 x 30 nm in size.*

Please see the response above and refer to Figure 6, Figure 7, and related figure supplements. While Reviewer #2 is correct that LGNRs are not widely used, we note that one feature of our manuscript is the first report of biodistribution data for these particles. While not in the scope of the current work, we hope that the use of LGNRs will become more widespread based on their advantages over regular GNRs in several respects, which we have recently reported.

*However, in the Discussion the authors specifically claim that their technique is applicable to many nanoparticle shapes "(for example, gold nanospheres, nanorods, nanocages, etc.)" […]. "ABIDE is capable of distinguishing such NPs from each other by spectral differences, enabling biodistribution studies of multiplexed NPs." This is not only a bold claim, which importantly is not supported by any data.*

Thank you to Reviewer #2 for this important point. Because we do not explicitly test gold nanocages or ex vivo spectral unmixing capabilities, we have revised these statements accordingly to reflect the particle types tested herein and the specific experiments we have performed. They now reflect our demonstrated abilities to characterize multiple particle types *in vivo* and to spectrally resolve different particle types within mixtures (although not yet demonstrated in tissue sections). Please refer to Figure 6, Figure 7, and their related figure supplements in the revised manuscript. Notably, the Nanoshells and LGNRs used in spectral unmixing tests exhibited similar near-infrared plasmonic peaks, yet they can be resolved from each other, likely as a result of differences in spectral width.

*The technique is only interesting enough for a broad audience if multiple different nanoparticle shapes such as spheres, cages, regular sized nanorods etc., can be analyzed with this method.*

We agree with Reviewer #2 that the ability to identify a number of nanoparticles is of key importance to a general audience. To address this, we have added extensive *ex vivo* analysis of gold nanoshells and spherical gold nanoparticles administered intravenously to mice. The particles we used are commercially-available Nanoshells (Nano Composix, San Diego, CA) and GNS@SiO_2_ (Oxonica, Mountain View, CA) that have been used in numerous applications to date (several are listed below). Please note that the GNS@SiO_2_ used in our experiments are currently under evaluation for potential clinical applications.

Examples of prior work using GNS@SiO_2_:

“Noninvasive molecular imaging of small living subjects using Raman spectroscopy” (Keren et al., PNAS 2008);

“A brain tumor molecular imaging strategy using a new triple-modality MRI-photoacoustic-Raman nanoparticle” (Kircher et al., Nat. Med. 2012);

“Affibody-functionalized gold-silica nanoparticles for Raman molecular imaging of the epidermal growth factor receptor” (Jokerst et al., Small 2011);

“Preclinical evaluation of Raman nanoparticle biodistribution for their potential use in clinical endoscopy imaging” (Zavaleta et al., Small 2011).

*And my concern is that the technique will run into issues with discriminating some of the other shapes from cellular components, as can already be predicted when considering the curves in "Figure 1—figure supplement 4". The large gold nanorods have a plasmonic peak in the near infrared, which may be much easier to discriminate by the machine-learning algorithm than for shapes such as e.g. spheres.*

*The authors need to show convincingly that this technique works for the other nanoparticle shapes as they claim.*

Reviewer #2 is correct that some particles may exhibit similar spectra to cellular components, a point that we acknowledged in our Discussion section. One way to circumvent this problem is to perform analysis on unstained tissue sections (in which case, two of the curves (for H&E stains) shown in Figure 1—figure supplement 4 would not be present). In fact, we found that it was important to analyze unstained tissue sections when detecting GNS@SiO_2_. While this requires consideration of sample preparation methods, we found that GNS@SiO_2_ detection and characterization in unstained tissues was as successful as our original detection of LGNRs, despite the ~300 nm difference in spectral peaks. Thus, scattering from (unstained) cellular components does not appear to preclude the detection of commonly-used nanoparticles that lack near infrared spectral signatures. We believe this new data provides a demonstration of the broader utility of our reported methods. Reviewer #2 is correct that spectra in the near-infrared will be easier to identify in stained sections. However, our results for GNS@SiO_2_ indicate that particles with visible spectrum signatures can also be studied with our methods, given the aforementioned preparation of unstained sections.

*Reviewer #3:*

*The paper presents methods and results of combining mathematical modeling, data analysis, hyper spectral imaging, related to imaging of nano materials with applications in cancer, angiogenesis, and others. I find this area really interesting, and the potential impact large, although tissue and cell physiology are not my area of expertise. Though I find the ideas very stimulating, the main issue that I have is that the mathematical modeling/data analysis used here is not very clear, and may not be near top shelf work. Combined with certain choices of validation expanded below, my understanding is that the contributions to imaging methodology is not sufficiently novel, nor inspire a lot of confidence that results are as best as possible. See below for detailed comments.*

Thank you to Reviewer #3 for the positive comments on our work. We hope that the revisions made to our manuscript and the responses below provide an adequate address of the reviewer’s concerns. As a general point, we acknowledge that the novelty of our work does not lie exclusively within a major advance in the field of machine learning. Rather, we believe our work’s novelty and (perhaps more importantly) utility resides in the combination of: 1) using machine learning to improve the state of biomedical hyperspectral imaging, 2) the specific biological insights and new data reported for nanoparticle microbiodistribution, and 3) the general biomedical capabilities developed for future applications (i.e., high-resolution biodistribution, studies of molecular targeting for therapy and imaging agent evaluation, etc.).

Furthermore, it is important to note that k-means clustering and nearest neighbor classification are only two parts of the full algorithm, which also includes defining an adaptive signal intensity threshold and pre-processing of the data. We believe that the full approach is novel and it is what makes ABIDE invariant to image acquisition parameters and applicable in many cases.

For more detail regarding novelty, please refer to the responses to Reviewer #1. Regarding whether our results are the best possible for machine learning methods, please refer to the detailed responses below.

*Regarding modeling and data analysis, a few questions. The topic of discerning the contributions of different elements (in this case nano materials versus others such as Eosin, Hematoxylin, etc.) from spectral measurements is a well studied one. While I understand the intricacies of this imaging experiment are not the same as other more well studied spectral unmoving problems, I'm not convinced that the wealth of other methods for linear and nonlinear unmixing don't apply here. In the paper I did not find any discussion related to this. Classification methods of the type authors claim to have attempted (nearest neighbor, SVMs, etc.) could be applied on such unmixed data, and to me this would constitute a more standard way of doing things. As is, the methodology regarding data analysis coupled to imaging does not seem very novel, and it is not presented in a way that can be related to many other already proposed methods to other seemingly similar applications.*

As part of the effort to detect the LGNRs in stained tissue sections, we explored various machine learning approaches. We actually started with supervised learning approaches, such as SVM and logistic regression, but then we found that k-means, when trained on slides of stained tissue that was injected with LGNRs, performed much better in detecting the LGNRs. This is because of the un-supervised nature of k-means, which automatically found the correct target LGNR spectrum, which turned out to be slightly different in tissue sections compared to the spectra of LGNRs dispersed on a glass slide. These results are explained in a report, which we referenced in our manuscript (Liba and Shaviv, 2014 in the revised manuscript), and have also been uploaded along with the revision.

An additional advantage of k-means as an unsupervised learning method is that it does not require pre-labeling the samples, which may be tedious. Rather, the researchers run the algorithm and choose the cluster that best fits their target spectrum (as determined through independent measurements such as Vis-NIR spectrometry).

We did not add the information regarding the different machine learning approaches in order to help the reader focus on the biomedical application, as we assumed the majority of readers will come from the application field and not from the computational side. If the reviewer believes that this information will help the readers, we are happy to include the comparison of machine learning methods as a supplementary document.

Furthermore, we believe that the ease of use of k-means and its popularity will encourage researchers to apply ABIDE for their research, while more complicated methods, such as deep-learning or unmixing, would make this work out of reach for researches who are not working in this field (k-means is more readily applied). We are hopeful that researchers who are familiar with or would like to try other methods, would be able to easily do so in post processing.

It may very well be that k-means is not the optimal algorithm to use for all applications, but we found that it is a great place to start in terms of result quality and diagnostic evaluation (particularly with respect to current methods for hyperspectral analysis, such as spectral angle mapping, described in the response to Reviewer #1). As in many machine-learning projects, it may be that the best algorithm would be tailored to the specific problem. For this demonstration, we sought to provide a generalizable method that will be broadly applicable and accessible for researchers working with various samples, particle types, and experimental designs.

As a note to the reviewer, we have also attempted linear unmixing methods and independent component analysis (ICA), however these did not produce qualitatively better results than k-means and therefore we did not perform a rigorous analysis on them. We believe they did not work well because the LGNR spectral component is not sufficiently independent from the tissue spectral components, and especially spectra that result from chromatic aberrations (this is discussed in detail within our manuscript’s results and methods sections). However, we are continuing to explore the potential use of linear and nonlinear unmixing for hyperspectral image analysis.

*Regarding the data analysis for validation, a couple of things seemed unclear to me. For the detection of false positives and false negatives, were these evaluations performed separately? And if so, is the validation criterion computed by checking whether there is one single pixel indicating nonmaterial (in case of false positives) present? Similar comments for the reverse situation (false negatives). It would be necessary to know the specifics of these details better, and even better have the data (e.g. histograms and if pixel counts are used, how thresholds are utilized). Also, how is the user defined parameter (it seems a manually selected threshold is used to initially determine if a pixel is potentially LGNR) handled during validation? Is data used in the validation stage also used by the user when selecting this parameter? It is a little confusing when in the subsection “Data processing and automatic biodistribution detection” the authors comment "We then manually (and in a manner blind to algorithmic classification)…". How is this possible given that for a positive detection the pixel must be classified as potential LGNR first? Am I missing something, or is it possible that authors are mixing training and testing data?*

First we would like to reassure that the training data sets were completely separate from the testing data sets. Training (meaning, calculating the clusters with k-means) was performed on images of LGNR injected tissue while testing was performed on different images and is detailed below.

The calculation of false positives and false negatives was done in two separate stages:

False negatives and true positives were calculated on images which include pure LGNRs only. The false negatives are the number of pixels (above the algorithm-defined threshold) that do not classify as LGNRs. Please see Figure 2—figure supplement 5 for an example of ABIDE detection in a pure LGNR sample, as well as the methods section describing this calculation.

False positives and true negatives were calculated on images, which include tissue only (i.e., no LGNRs present). The false positives are the number of pixels (above the algorithm-defined threshold) that classify as LGNRs. Please refer to Figure 2—figure supplement 4 for qualitative visualization of typical false positive detection. Also, Figure 1—figure supplement 2 shows an example histogram for a tissue-only sample with annotations depicting the threshold values (minHist = noise threshold and peakHist = particle threshold) determined by the algorithm for each image.

In addition to that, we performed another validation of the algorithm by testing it on slides of tissue that were injected with LGNRs (4 separate sections of kidney from LGNR-injected animals were used for this purpose). In that case, we observed and manually classified the spectra of >200 randomly-selected pixels (>50 pixels per each of the 4 FOVs) and compared our observation with the results of the algorithm (we were independent and blind to the results of the algorithm during this process, as was mentioned in line 411 of our initial submission). This analysis provided a measurement of FP, TP, FN and TN. This test was done on different images than the training data.

We note that the raw data for sensitivity and specificity measurements were provided as ancillary text documents in the original submission. Following the reviewers comment, we have included the numbers of measurements (pixel counts) in each category alongside the ratios previously provided. These raw data have been added as Figure 2—figure supplement 1.

An adaptive threshold was calculated and used before the classification process. The adaptive threshold is a result of the algorithm described in the paper (section “Data processing and automatic biodistribution detection” and Figure 1—figure supplement 2), and has no user-defined parameters, other than “sanity parameters” ensuring the threshold is in a certain range. The algorithm, which determines the threshold, was qualitatively validated on over 20 images during its development. In the validation process we manually validated that the pixels that have enough intensity to be LGNRs fall above the LGNR threshold and that all tissue falls above the tissue threshold. Furthermore, we manually validated these “threshold maps” (such as the one in Figure 1—figure supplement 2) for all the images used in the manuscript, and in general for our research.

To further clarify, only pixels above the image-adaptive LGNR threshold were considered for classification purposes. Only these pixels appear in the calculation of TP, FP, TN, FN.

*A few more comments below:*

*The Results section reads more like a Methods section. If format is to be followed, I'd suggest only describing results in the Results section.*

Thank you to Reviewer #3 for this feedback. We wanted to ensure that readers could readily follow the logic of key experiments and their results, and we felt that including some key aspects of our methodology was relevant to this aim. More detailed methods needed for reproducibility were avoided in the Results section and instead provided in the Methods section.

*Results, second paragraph: K-means is most commonly referred as a clustering algorithm. Once clusters are identified, one can use this information to design a multitude of classification methods, but authors must specify which they used. Presumably the simplest would be the nearest neighbor cluster center. Are the cluster centers utilized as ground truth?*

We thank the reviewer for this clarification. Indeed we used a nearest centroid (or nearest neighbor) classifier, based on the Euclidean distance to the cluster centers. We have now updated the manuscript to clarify this aspect of the algorithm.

We are not sure what the reviewer means by “ground truth” in this context. In the sense that each spectral cluster of interest can be cross-validated with spectral measurements from other techniques (i.e., Vis-NIR spectrometry, HSM of pure particles), the clusters do constitute a ground truth (as in the real, empirically assayed particle or dye spectra).

[Editors’ note: the author responses to the re-review follow.]

*The manuscript has been improved but there are some remaining issues that need to be addressed before acceptance. They mainly concern the tendency of 'over-selling' some of the approaches and accomplishments of the work. We especially encourage you to reconsider the use of an acronym for this technology. We are concerned about the use of acronyms as cheap eye-catchers. Moreover, a new expert in machine learning has been brought to the panel because previous Reviewer #3 was unavailable. Given the rapid pace of the machine learning field it seems inappropriate to call the application of a clustering method (and even a very classic one, which is taught in undergraduate computer science course) 'machine learning'. It is perfectly fine to use standard methods if they solve the task, but there is a discomfort among reviewers that machine learning is merely used as a catchy phrase in this case. This could fire back on your work. We thus encourage you to revise some of the language in your manuscript, including the Abstract, and to address the few other comments listed below.*

We would like to thank the editors and reviewers for their favorable consideration and constructive comments on our work. The newest version of our manuscript has addressed the outstanding concerns as described in the point-by-point response to reviewer comments provided below. In brief:

The title and Abstract have been revised to avoid over-selling.

The acronym “ABIDE” (Automated Biodistribution Detection) has been replaced with the abbreviation “HSM-AD” (Hyperspectral Microscopy with Adaptive Detection). This is a modification of the existing abbreviation “HSM.” We believe the abbreviation “HSM-AD” is more descriptive of our method and should relieve the concern about eye-catcher acronyms. Generally, we note that some form of abbreviation is necessary to aid readability.

The phrase “machine learning” has been replaced throughout the manuscript with more specific language that describes the relevant components of the algorithm. These changes should avoid misleading readers.

*Reviewer #1:*

*I've found that the authors’ comments are very appropriate and to the point even though in some cases I'm not too familiar with some of the points discussed.*

We thank Reviewer #1 for the positive consideration of our revised manuscript.

*My major concern is always related to my original comment. Previous similar work from Brenner using a basically identical setup hardware is already present in the literature. Because the scientific contribution of the submitted paper consists in proposing an analysis procedure based on a machine learning algorithm with the intent of extending the work of Brenner and co-workers and others, it is critical to determine the novelty of the proposed algorithm.*

We wish to emphasize the advances of our work with respect to previously published studies including the work by Brenner, as noted in extensive detail within our original point-by-point response document. Summarily, the novel aspects of the algorithm relative to existing work include:

a) The first use of hyperspectral imaging for a full systemic biodistribution study;

b) Single-particle detection sensitivity in tissues;

c) The demonstration of a fully adaptive (i.e., corrects for variable acquisition parameters and illumination) and automated (i.e., notably higher throughput image analysis) method for qualitative and quantitative assessments of biological samples using hyperspectral imaging;

d) This method can provide diagnostic values to aid image interpretation. Moreover, the detection method itself achieves high sensitivity and high specificity.

We also note the novel aspects of the experimental data obtained using the algorithm package:

a) Sub-organ localization patterns of three nanoparticle types with unique shapes, spectra, and sizes. Numerous insights are provided regarding the influence of size and surface coating on differences in both intra- and inter-organ biodistribution. For spherical gold particles, the hyperspectral imaging results correlated well with biodistribution data we obtained using ICP-MS;

b) The first use of hyperspectral microscopy to detect molecularly-targeted nanoparticle uptake in tumors.

The algorithm’s high-throughput and ease of use offer notable practical advances in hyperspectral analysis. We believe there is significant value in this sense because it enables a wide host of studies that rely on the analysis of large imaging datasets (for example, systemic biodistribution studies).

*Overall I found the data provided by the authors very compelling and particularly interesting.*

Again, thank you to Reviewer #1 for helpful and positive comments throughout the review process.

*Reviewer #2:*

*The authors are now showing feasibility data from two other nanoparticle shapes, and have therefore satisfactorily addressed my previous main concern whether ABIDE may be versatile enough with regards to different nanoparticle shapes and sizes.*

Thank you to Reviewer #2. We are pleased that these additions have addressed the reviewer’s original concerns.

*There are a few remaining issues.*

*General comment: The revised version does not include any tracked changes or other markings to indicate where changes were made, which made the review of the paper quite difficult. I am making this comment not because I want to review another version with track changes, but to make it clear that this limitation may have reduced my ability to catch all remaining or new issues.*

We thank the reviewer for the concern. Please note that we did in fact include two versions of the manuscript during the first round of revisions: one with tracked changes and one without. The tracked changes version was included as manuscript item #13, entitled: “Document with markup showing changes from original full submission to the revised manuscript.”

*Specific comments:*

1) In the PowerPoint slides the authors provided in their rebuttal (for the reviewers only), it says "Example of new in vivo data". I find this misleading, as I could not find any data in the entire paper that was acquired "in vivo". A reviewer who does not carefully examine the manuscript may be misled by the rebuttal summary slides that this is in fact all acquired in vivo and not catch this discrepancy between the summary and the actual paper. I am not expecting the authors to provide true in vivo data, but would like to clarify what the authors meant by that.

Thank you to Reviewer #2 for catching this error. Because we are analyzing tissue sections, all data are necessarily ex vivo. We have ensured that this is accurately reflected throughout the manuscript.

*2) The title of the paper is overstated with regards to claiming "single particle sensitivity" and this needs to be changed or else would be misleading. Figure 2—figure supplement 8 is the only figure that shows any data that would support that, and only in the very large gold nanorods (which by some definitions would not represent a nanoparticle). "Likely" as is stated in the figure legend is probably an honest assessment by the authors, but not enough to make such a major claim, and there is no evidence for this to work in the other two nanoparticles that are now included. I suggest replacing "…with single particle sensitivity" with "… in tissue sections" or "… in histological slices".*

We have revised the title pursuant to Reviewer #2’s request. The new title is: “A hyperspectral method to assay the microphysiological fates of nanomaterials in histological samples.”

*I would be willing to accept the manuscript pending these clarifications/changes if the other reviewers agree that their areas of expertise were addressed sufficiently as well.*

Thank you to Reviewer #2 for the helpful comments and positive consideration of our work throughout the review process.

*Reviewer #4:*

*This revised manuscript describes an interesting and straightforward development of an approach for detecting nanoparticles in hyperspectral dark field images. It has been significantly improved based on comments in the initial reviews. The results presented demonstrate an impressive ability to detect and quantify these nanoparticles in tissue images.*

We thank the reviewer for the positive consideration of our work.

*From an image processing/analysis/machine learning point of view, the approach is not novel or instructive (clustering spectra, especially with manual tuning, barely qualifies to be called machine learning, and does not seem to warrant a new acronym).*

We agree that the method we have used for clustering and classification is not novel and is indeed widely used. The novelty of our work lies in the application of these methods for the detection of particles and obtaining their biodistribution using hyperspectral imaging, which has not been demonstrated previously.

Please note that the described method, which we refer to as HSM-AD (previously “ABIDE”), includes two auxiliary algorithms in addition to clustering and classification: adaptive thresholding (Figure 1—figure supplement 2) and pre-processing of the hyperspectral signals (Figure 1—figure supplement 3). The combination of these methods make HSM-AD non-trivial and, most importantly, critical for the automated and generalizable detection of particles in tissue samples.

Following the reviewers comment, we have removed the use of the term “machine-learning” from the manuscript. Please refer to the document with tracked changes to see these modifications.

We believe that using an abbreviation for our method will improve readability. Therefore, we have changed the acronym “ABIDE” to “HSM-AD” (Hyperspectral Microscopy with Adaptive Detection) to avoid overly catchy terms. Please note that HSM is not a new acronym but rather an existing abbreviation.

*Hence the significance of the manuscript must derive from the future importance of the method's application, something this reviewer is not qualified to judge.*

We note that there are several significant aspects of the current manuscript (described in the response to Reviewer #1, the point-by-point response from the first round of revisions, and in the conclusion section of the main text). For example, we were able to characterize unique patterns of particle accumulation within liver, kidney, and spleen tissue that have not previously been reported.

Reviewer #4 correctly notes that the manuscript should be relevant for future applications. We believe that our method can be used in the future for rigorous preclinical assessments of nanoparticles intended for diagnostic or therapeutic uses. As a specific example, we are currently using our method to evaluate the uptake and clearance of a clinical nanoparticle candidate (GNS@SiO_2_) following intravenous versus oral administration routes. The method’s high sensitivity, specificity, and imaging resolution provide particle biodistribution data with greater detail than the data obtained from existing techniques like ICP (see Figure 7 and related supplements for intravenous data from these particles). For example, our method can be used to evaluate the extent of elimination of particles from a tissue of interest over time, which is critical for preclinical evaluations by the FDA and other regulatory agencies.